# DOUBLY ROBUST INSTANCE-REWEIGHTED ADVERSARIAL TRAINING

**Daouda A. Sow**
Department of ECE
The Ohio State University
sow.53@osu.edu

**Sen Lin**
Department of CS
University of Houston
slin50@central.uh.edu

**Zhangyang Wang**
Visual Informatics Group
University of Texas at Austin
atlaswang@utexas.edu

**Yingbin Liang**
Department of ECE
The Ohio State University
liang.889@osu.edu

## ABSTRACT

Assigning importance weights to adversarial data has achieved great success in training adversarially robust networks under limited model capacity. However, existing instance-reweighted adversarial training (AT) methods heavily depend on heuristics and/or geometric interpretations to determine those importance weights, making these algorithms lack rigorous theoretical justification/guarantee. Moreover, recent research has shown that adversarial training suffers from a severe non-uniform robust performance across the training distribution, e.g., data points belonging to some classes can be much more vulnerable to adversarial attacks than others. To address both issues, in this paper, we propose a novel doubly-robust instance reweighted AT framework, which allows to obtain the importance weights via exploring distributionally robust optimization (DRO) techniques, and at the same time boosts the robustness on the most vulnerable examples. In particular, our importance weights are obtained by optimizing the KL-divergence regularized loss function, which allows us to devise new algorithms with a theoretical convergence guarantee. Experiments on standard classification datasets demonstrate that our proposed approach outperforms related state-of-the-art baseline methods in terms of average robust performance, and at the same time improves the robustness against attacks on the weakest data points.

## 1 INTRODUCTION

Deep learning models are known to be vulnerable to malicious adversarial attacks Nguyen et al. (2015), i.e., small perturbation added to natural input data can easily fool state-of-the-art networks. Given that these deep neural networks are being heavily deployed in real-life applications, even in safety-critical applications, adversarial training (AT) Madry et al. (2017); Athalye et al. (2018a); Carmon et al. (2019) has been proposed for training networks to be robust to adversarial attacks Athalye et al. (2018b); Szegedy et al. (2013); Goodfellow et al. (2014); Papernot et al. (2016); Nguyen et al. (2015); Zhang et al. (2021b; 2020a). In particular, most existing defense strategies are based on the recipes similar to AT Madry et al. (2017), where the goal is to minimize the average loss of the worst-case adversarial data for the training distribution via solving a minimax optimization problem.

Despite its success, the traditional AT method Madry et al. (2017) has some major limitations. First, even though existing overparameterized neural networks seem to be good enough for natural data, highly adversarial data consumes much more model capacity compared to their clean counterpart, making the minimization of the **uniform** average adversarial loss a very pessimistic goal, as argued in Zhang et al. (2020b). To overcome this limitation, recent works Zhang et al. (2020b); Liu et al. (2021a); Zeng et al. (2021); Ding et al. (2018) assign an **importance weight** to each data point in the training distribution, in order to emphasize the ones that are critical to determining the model's decision boundaries. By allowing more careful exploitation of the limited model capacity, such a simple **instance-reweighted** scheme combined with traditional adversarial training has yielded a

significant boost in the robust performance of current adversarially trained models. Yet, existing methods for instance-reweighted AT mostly adopt heuristic techniques and/or geometric intuitions in order to compute the instance weights, which makes these algorithms lack a principled and rigorous theoretical justification/guarantee. This hence motivates the following question we ask:

*How to systematically determine the importance weights via a principled approach, rather than resorting to heuristics/interpretations which are often sub-optimal?*

Moreover, as observed in Tian et al. (2021), another critical limitation of the transitional AT method is that it suffers a severe *non-uniform* performance across the empirical distribution. For example, while the average robust performance of the AT method on the CIFAR10 dataset can be as high as $49\%$, the robust accuracy for the weakest class is as low as $14\%$, which depicts a huge disparity in robust performance across different classes. We note that such a non-uniform performance across classes is also slightly observed in the standard training with clean data, but its severity is much worsened in adversarial training (see Figure 1). Indeed, this is a critical limitation that requires special attention as, in a real-world situation, a more intelligent attacker can, in fact, decide which examples to attack so as to achieve a much higher success rate (e.g., $86\%$ when attacking the most vulnerable class). This non-uniform robust performance is even worsened in the case of imbalanced training distributions Wu et al. (2021); Wang et al. (2022), where the robust performance for the most vulnerable class can be as low as $0\%$. This motivates our second question given below:

*Can such an issue of non-uniform performance particularly over imbalanced datasets be addressed at the instance level simultaneously as we design the importance weights to address the first question?*

In this paper, we propose a novel doubly robust instance reweighted optimization approach to address both of the above questions.

## 1.1 OUR CONTRIBUTIONS

**(A novel principled framework for instance reweighted AT)** In order to determine the instance weights for AT in a theoretically grounded way, we propose a novel doubly robust instance reweighted optimization framework, based on distributionally robust optimization (DRO) Rahimian & Mehrotra (2019); Qian et al. (2019) and bilevel optimization (Zhang et al., 2022; Pedregosa, 2016; Grazzi et al., 2020b). Through building a model that is robust not only to the adversarial attacks but also to the worst-case instance weight selections, our framework (a) enjoys better robust performance than existing instance-reweighted schemes based on heuristic/geometric techniques Zhang et al. (2020b); Liu et al. (2021a); Zeng et al. (2021) as well as tradtional AT baselines Madry et al. (2017); and (b) addresses the non-uniform issues Tian et al. (2021); Pethick et al. (2023) of traditional AT by carefully optimizing the instance weights so as to boost the robust performance of the most vulnerable examples. Moreover, the proposed framework can be reformulated into a new finite-sum compositional bilevel optimization problem (CBO), which can be of great interest to the optimization community on its own.

**(A novel algorithm with theoretical guarantee)** Solving the proposed doubly robust optimization problem is technically challenging, including the non-differentiability of the optimizer for the constrained inner level problem and the biased hypergradient estimation for the compositional outer level problem. To tackle these challenges, we first propose a penalized reformulation based on the log-barrier penalty method, and then develop a novel algorithm which exploits the implicit function theorem and keeps track of a running average of the outer level composed function values. Our algorithm not only leads to a robust model for the proposed instance reweighted optimization problem but also provides a solution to the generic compositional bilevel optimization problem. Under widely adopted assumptions in the bilevel (Grazzi et al., 2020a; Ji et al., 2021; Rajeswaran et al., 2019; Ji & Liang, 2021) and compositional optimization Wang et al. (2017); Chen et al. (2021b); Lian et al. (2017); Blanchet et al. (2017); Devraj & Chen (2019) literature, we further establish the convergence guarantee for the proposed algorithm.

**(Strong experimental performance)** Experiments on several balanced and imbalanced image recognition datasets demonstrate the effectiveness of our proposed approach. In particular, on CIFAR10 our approach yields +3.5% improvement in overall robustness against PGD attacks Madry et al. (2017) with most of it coming from boosting robustness on vulnerable data points.

## 1.2 RELATED WORK

**Adversarial training for robust learning** Adversarial training (AT) Madry et al. (2017); Athalye et al. (2018a); Carmon et al. (2019) was proposed for training deep neural networks robust to malicious adversarial attacks Goodfellow et al. (2014); Tramèr et al. (2017). In particular, Madry et al. (2017) introduced a generic AT framework based on minimax optimization with the goal of minimizing the training loss of the worst-case adversarial data for the training distribution. However, despite AT method being still considered as one of the most powerful defense strategies, Rice et al. (2020) highlights a severe decrease in robust performance of traditional AT when training is not stopped early, a phenomenon they dubbed *robust overfitting*. Several extensions of the standard AT method have been proposed to mitigate this intriguing problem, such as data augmentation-based techniques Rebuffi et al. (2021); Gowal et al. (2021), or smoothing-based methods Chen et al. (2021a); Yang et al. (2020a;b). Zhang et al. (2019) proposed a theoretically grounded objective for AT to strike a balance between robust and natural performance. However, those methods suffer a severe non-uniform performance across classification categories, as observed in Tian et al. (2021). Our proposed framework helps mitigate this drawback by carefully optimizing for the most vulnerable data points.

**Instance reweighted adversarial training** Another line of works Zhang et al. (2020b); Liu et al. (2021a); Zeng et al. (2021); Ding et al. (2018) assign an importance weight to each data point in the empirical distribution and minimize the weighted adversarial losses. This has been shown to significantly boost the performance of AT due to more careful exploitation of the limited capacity of large deep neural networks to fit highly adversarial data, and helps overcome robust overfitting to some extent Zhang et al. (2020b). For example, in the geometry-aware adversarial instance reweighted adversarial training (GAIRAT) Zhang et al. (2020b) method, the instance weight is computed based on the minimum number of PGD Madry et al. (2017) steps required to generate a mis-classified adversarial example. Liu et al. (2021a) leverages probabilistic margins to compute weights. Existing approaches for instance reweighted AT are, however, all based on heuristics/geometric intuitions to determine the weights. In this paper, we propose a principled approach to instance-reweighted AT by exploiting robust optimization techniques Qian et al. (2019); Rahimian & Mehrotra (2019).

Instance reweighting has also been used in the context of domain adaptation Jiang & Zhai (2007), data augmentation Yi et al. (2021), and imbalanced classification Ren et al. (2018). By determining the instance weights in a more principled way, our method also has the potential to be applied to these contexts, which we leave as future work.

Due to space limitation, more discussions about related literature in Bilevel Optimization and Stochastic Compositional Optimization is deferred to Appendix A.

## 2 PRELIMINARY ON AT

**Traditional AT.** The traditional adversarial training (AT) Madry et al. (2017) framework is formulated as the following minimax optimization problem over the training dataset $\mathcal{D} = \{(x_i, y_i)\}_{i=1}^{M}$

$$\min_{\theta} \frac{1}{M} \sum_{i=1}^{M} \max_{\delta \in \mathcal{C}} \ell(x_i + \delta, y_i; \theta), \tag{1}$$

where $\ell(x_i + \delta, y_i; \theta)$ is the loss function on the adversarial input $x_i + \delta$, $\mathcal{C}$ is the treat model that defines the constraint on the adversarial noise $\delta$, and $\theta \in \mathbb{R}^d$ corresponds to the model parameters. Thus, the traditional AT builds robust models by optimizing the parameters $\theta$ for the average worst-case adversarial loss $\ell(x_i + \delta, y_i; \theta)$ over the training dataset $\mathcal{D}$. A natural solver for the problem in Equation (1) is the AT algorithm Madry et al. (2017), where 1) the projected gradient descent (PGD) Madry et al. (2017) method is first adopted to approximate the worst-case adversarial noise $\delta$ and 2) an outer minimization step is performed on the parameters $\theta$ using stochastic gradient descent (SGD) methods. However, the traditional AT is known to consume tremendous amount of model capacity due to its overwhelming smoothing effect of natural data neighborhoods Zhang et al. (2020b). In other words, the traditional AT robustifies models by making decision boundaries far away from natural data points so that their adversarial counterparts are still correctly classified (i.e., do not cross the decision boundary), and thus requires significantly more model capacity compared to the standard training on clean data.

**Instance Reweighted AT.** The geometry-aware approach in Zhang et al. (2020b) introduces a new line of methods that **reweights** the adversarial loss on each individual data point in order to address

the drawback of traditional AT. The key motivation is that distinct data points are unequal by nature and should be treated differently based on how important they participate on the selection of decision boundaries. Hence, the learning objective of the geometry-aware instance-reweighted adversarial training (GAIRAT) method as well as its variants Zhang et al. (2020b); Liu et al. (2021a); Zeng et al. (2021) can be written as

$$\min_{\theta} \sum_{i=1}^{M} w_i \max_{\delta \in \mathcal{C}_i} \ell(x_i + \delta, y_i; \theta) \quad \text{with} \quad \sum_{i=1}^{M} w_i = 1 \text{ and } w_i \geq 0, \quad (2)$$

where the constraints on the weights vector $w = (w_1, ..., w_M)^\top$ are imposed in order to make Equation (2) consistent with the original objective in Equation (1). This framework assumes that the weight vector $w = (w_1, ..., w_M)^\top$ can be obtained separately and the goal is only to optimize for $\theta$ once an off-the-shelf technique/heuristic can be used to compute $w$. Intuitively, the key idea driving the weight assignments in instance reweighted methods is that larger weights should be assigned to the training examples closer to the decision boundaries, whereas the ones that are far away should have smaller weights because they are less important in determining the boundaries. The major difference among the existing instance reweighted AT methods lies in the heuristics used to design/compute the instance weights $w_i, i = 1, ..., M$. However, none of those methods adopt a scheme that is theoretically grounded, nor does the formulation in Equation (2) provide a way of determining those weights.

**Bilevel Optimization Formulation for AT.** Along a different line, bilevel optimization has recently been leveraged to develop a more powerful framework for adversarial training Zhang et al. (2022):

$$\min_{\theta} \frac{1}{M} \sum_{i=1}^{M} \ell(x_i + \delta_i^*(\theta), y_i; \theta) \quad \text{s.t.} \quad \delta_i^*(\theta) = \arg\min_{\delta \in \mathcal{C}_i} \ell'(x_i + \delta, y_i; \theta), \quad (3)$$

where for each data point $(x_i, y_i)$, $\delta_i^*(\theta)$ represents some worst-case/optimal adversarial noise under the attack loss function $\ell'(\cdot; \theta)$. Such a bilevel optimization formulation of AT has key advantages over the traditional framework in Equation (1). First, the traditional AT can be recovered by setting the attack objective to be the negative of the training objective, i.e., $\ell'(\cdot; \theta) = -\ell(\cdot; \theta)$. Second, the bilevel formulation gives one the flexibility to separately design the inner and outer level objectives, $\ell'$ and $\ell$, respectively. These key advantages make the formulation in Equation (3) a more generic and powerful framework than the one in Equation (1). As we will see next, this enables us to independently construct a new outer level objective that also solves for the instance weights $w$, and an inner level objective for regularized attack.

## 3 Proposed Framework for Instance Reweighted AT

### 3.1 DONE: Doubly Robust Instance Reweighted AT

Using the bilevel formulation for AT in Eq. equation 3, we can incorporate the instance reweighted idea as

$$\min_{\theta} \sum_{i=1}^{M} w_i \ell(x_i + \delta_i^*(\theta), y_i; \theta) \text{ s.t. } \delta_i^*(\theta) = \arg\min_{\delta \in \mathcal{C}_i} \ell'(x_i + \delta, y_i; \theta) \text{ with } \sum_{i=1}^{M} w_i = 1 \text{ and } w_i \geq 0. \quad (4)$$

Based on bilevel optimization and distributionally robust optimization (DRO), we next propose a new framework for AT which determines the weights $w$ in a more principled way rather than using heuristic methods. Specifically, by letting $w$ maximize the weighted sum of the adversarial losses $\ell(x_i + \delta_i^*(\theta), y_i; \theta), i = 1, ..., M$, we seek to build a model in the outer level problem that is robust not only to the adversarial attacks but also to the worst-case attack distribution:

$$\min_{\theta} \max_{w \in \mathcal{P}} \sum_{i=1}^{M} w_i \ell(x_i + \delta_i^*(\theta), y_i; \theta) - r \sum_{i=1}^{M} w_i \log(M w_i) \quad \text{s.t.} \quad \delta_i^*(\theta) = \arg\min_{\delta \in \mathcal{C}_i} \ell'(x_i + \delta, y_i; \theta), \quad (5)$$

where $\mathcal{P}$ represents the probability simplex, i.e., $\mathcal{P} = \{w \in \mathbb{R}^M : \sum_{i=1}^{M} w_i = 1 \text{ and } w_i \geq 0\}$, and the term $r \sum_{i=1}^{M} w_i \log(M w_i)$ in the outer level objective captures the KL-divergence between $w$ and the uniform weight distribution, which is a widely adopted choice of regularizer in the DRO literature Rahimian & Mehrotra (2019). Note that the regularization parameter $r > 0$ controls the tradeoff between two extreme cases: 1) $r = 0$ leads to an un-regularized problem (as we comment

below), and 2) $r \to \infty$ yields $w_i \to \frac{1}{M}$, and hence, we recover the average objective in Equation (1). Such a regularizer is introduced to promote the balance between the uniform and worst-case weights $w$; otherwise the outer level objective in Equation (5) becomes linear in weights vector $w$, which makes the solution of the 'max' problem to be trivially a one-hot vector $w$ (where the only '1' is at index $i$ with the largest adversarial loss), and in practice, such a trivial one-hot vector $w$ makes the optimization routine unstable and usually hurts generalization to the training distribution Qian et al. (2019); Wang et al. (2021).

Overall, the formulation in Equation (5) becomes a **doubly robust** bilevel optimization: (a) the inner level finds the worst-case noise $\delta$ in order to make the model parameters $\theta$ robust to such adversarial perturbation of data input; and (b) the outer level finds the worst-case reweighting first so that the optimization over the model $\theta$ can focus on those data points with high loss values, i.e., the optimization over $\theta$ is over the worst-case adversarial losses.

## 3.2 AN EQUIVALENT COMPOSITIONAL BILEVEL OPTIMIZATION PROBLEM

An important consequence of choosing the KL-divergence as the regularizer is that the $\max$ problem in the outer objective of Equation (5) admits a unique solution $w^*(\theta)$ (see Qi et al. (2021) for proof), which has its $i$-the entry given by $w_i^*(\theta) = \exp\left(\frac{\ell_i(\theta, \delta_i^*(\theta))}{r}\right) / \sum_j \exp\left(\frac{\ell_j(\theta, \delta_j^*(\theta))}{r}\right)$. Here we denote $\ell_i(\theta, \delta_i^*(\theta)) = \ell(x_i + \delta_i^*(\theta), y_i; \theta)$. Substituting this optimal weights vector $w^*(\theta)$ back in Equation (5) yields the following equivalent optimization problem

$$\min_\theta r \log \left( \frac{1}{M} \sum_{i=1}^M \exp\left( \frac{\ell_i(\theta, \delta_i^*(\theta))}{r} \right) \right) \quad \text{s.t.} \quad \delta_i^*(\theta) = \arg\min_{\delta \in \mathcal{C}_i} \ell_i'(\theta, \delta). \quad (6)$$

Problem (6) is, in fact to the best of our knowledge, a novel optimization framework, which we define as a compositional bilevel optimization problem. Without the inner level problem, stochastic algorithms with known convergence behaviors have been devised for the single-level compositional problem. Nevertheless, directly solving problem (6) suffers from several key technical challenges. In particular, the fact that the minimizer of the inner level constrained problem in Equation (6) may not be differentiable w.r.t. to the model parameter $\theta$ prevents the usage of implicit differentiation for solving the bilevel optimization problem.

To tackle this challenge, we propose a penalized reformulation based on the log-barrier penalty method. More specifically, we consider $\ell_\infty$-norm based attack constraint given by $\mathcal{C} = \{\delta \in \mathbb{R}^p : \|\delta\|_\infty \le \epsilon, x + \delta \in [0, 1]^p\}$ for radius $\epsilon > 0$ and input $x \in \mathbb{R}^p$. In this case, the constraint set $\mathcal{C}$ can be written in the form of linear constraint $A\delta \le b$ with $A = \left(I_p, -I_p\right)^\top \in \mathbb{R}^{2p \times p}$ and $b = \left(\min(\epsilon\mathbf{1}_p, \mathbf{1}_p - x), \min(\epsilon\mathbf{1}_p, x)\right)^\top \in \mathbb{R}^{2p}$. With this, we can reformulate the inner problem in Equation (6) as $\delta_i^*(\theta) = \arg\min_{\{A_i \delta \le b_i\}} \ell_i'(\theta, \delta)$, where $A_i$ and $b_i$ are realizations of aforementioned $A$ and $b$ for input $x_i$. By using the log-barrier penalty method to penalize the linear constraint into the attack objective, the optimization problem (6) becomes

$$\min_\theta \mathcal{L}(\theta) \coloneqq r \log \left( \frac{1}{M} \sum_{i=1}^M \exp\left( \frac{\ell_i(\theta, \hat{\delta}_i^*(\theta))}{r} \right) \right) \quad \text{s.t.} \quad \hat{\delta}_i^*(\theta) = \arg\min_{\delta \in \mathcal{C}_i} \ell_i^{bar}(\theta, \delta), \quad (7)$$

where $\ell_i^{bar}(\theta, \delta) \coloneqq \ell_i'(\theta, \delta) - c \sum_{k=1}^{2p} \log(b_k - \delta^\top a_k)$, $a_k$ denotes the $k$-th row of matrix $A_i$ and $b_k$ is the $k$-th entry of vector $b_i$. Note that now the constraint $\{\delta \in \mathcal{C}_i\}$ is never binding in Equation (7), because the log-barrier penalty forces the minimizer of $\ell_i^{bar}(\theta, \delta)$ to be strictly inside the constraint set. Based on this, we show that the minimizer $\hat{\delta}_i^*(\theta)$ becomes differentiable, i.e., $\frac{\partial \hat{\delta}_i^*(\theta)}{\partial \theta}$ exists when $\ell_i'(\theta, \delta)$ is twice differentiable and under some mild conditions. With the smoothness of $\hat{\delta}_i^*(\theta)$, we also provide the expression of the gradient $\nabla \mathcal{L}(\theta)$ in the following proposition.

**Proposition 1.** *Let $\ell_i'(\theta, \delta)$ be twice differentiable. Define $\gamma_k = 1/(b_k - a_k^\top \hat{\delta}_i^*(\theta))^2$, $k = 1, ..., 2p$ and diagonal matrix $C_i(\theta) = c \, \mathrm{diag}\left(\gamma_1 + \gamma_{p+1}, \gamma_2 + \gamma_{p+2}, ..., \gamma_p + \gamma_{2p}\right)$. If $\nabla_\delta^2 \ell_i'(\theta, \hat{\delta}_i^*(\theta)) + C_i(\theta)$ is invertible, then the implicit gradient $\frac{\partial \hat{\delta}_i^*(\theta)}{\partial \theta}$ exists and we have*

$$\nabla \mathcal{L}(\theta) = \frac{r \sum_{i=1}^M \left( \nabla_\theta \, g_i(\theta, \hat{\delta}_i^*(\theta)) - \nabla_{\theta\delta} \, \ell_i'(\theta, \hat{\delta}_i^*(\theta)) \left[\nabla_\delta^2 \, \ell_i'(\theta, \hat{\delta}_i^*(\theta)) + C_i(\theta)\right]^{-1} \nabla_\delta \, g_i(\theta, \hat{\delta}_i^*(\theta)) \right)}{\sum_{i=1}^M g_i(\theta, \hat{\delta}_i^*(\theta))},$$

*where $g_i(\theta, \hat{\delta}_i^*(\theta)) = \exp\left( \frac{\ell_i(\theta, \hat{\delta}_i^*(\theta))}{r} \right)$.*

Proposition 1 provides the expression of the total gradient $\nabla \mathcal{L}(\theta)$, which is useful for practical implementation of implicit differentiation based algorithms for problem (6). Moreover, as in Zhang et al. (2022), when $\ell_i'(\theta, \cdot)$ is modeled by a ReLU-based deep neural network, the hessian $\nabla_\delta^2 \ell_i'(\theta, \delta)$ w.r.t. input $\delta$ can be safely neglected due to the fact that ReLU network generally lead to piece-wise linear decision boundaries w.r.t. its inputs Moosavi-Dezfooli et al. (2019); Alfarra et al. (2022), i.e., $\nabla_\delta^2 \ell_i'(\theta, \delta) \approx 0$. Further, the diagonal matrix $C_i(\theta)$ can be efficiently inverted. Hence, in order to approximate $\nabla \mathcal{L}(\theta)$, we only need Jacobian-vector product computations which can be efficiently computed using existing automatic differentiation packages.

## 3.3 COMPOSITIONAL IMPLICIT DIFFERENTIATION (CID)

To solve our reformulated problem (7) for AT, we consider the following generic compositional bilevel optimization problem, which can be of great interest to the optimization community:

$$\min_\theta F(\theta) := f\left(g\left(\theta, \delta^*(\theta)\right)\right) = f\left(\frac{1}{M} \sum_{i=1}^{M} g_i\left(\theta, \delta_i^*(\theta)\right)\right) \tag{8}$$

$$\text{s.t. } \delta^*(\theta) = (\delta_1^*(\theta), ..., \delta_M^*(\theta)) = \underset{(\delta_1, ..., \delta_M) \in \mathcal{V}_1 \times ... \times \mathcal{V}_M}{\arg\min} \frac{1}{M} \sum_{i=1}^{M} h_i\left(\theta, \delta_i\right),$$

which can immediately recover problem (7) by setting $g_i = \exp\left(\frac{\ell_i(\theta, \hat{\delta}_i^*(\theta))}{r}\right)$, $h_i = \ell_i'(\theta, \delta) - c \sum_{k=1}^{2p} \log(b_k - \delta^\top a_k)$, and the constraint set $\mathcal{V}_i = \mathcal{C}_i$. Here the outer functions $g_i(\theta, \delta) : \mathbb{R}^d \times \mathbb{R}^p \to \mathbb{R}^m$ and $f(z) : \mathbb{R}^m \to \mathbb{R}$ are generic nonconvex and continuously differentiable functions. The inner function $h_i(\theta, \delta) : \mathbb{R}^d \times \mathcal{V}_i \to \mathbb{R}$ is a twice differentiable and admits a unique minimizer in $\delta$, $\mathcal{V}_i$ is a convex subset of $\mathbb{R}^p$ that is assumed to contain the minizers $\delta_i^*(\theta)$. We collect all inner loop minimizers into a single vector $\delta^*(\theta)$. The goal is to minimize the total objective function $F(\theta) : \mathbb{R}^d \longrightarrow \mathbb{R}$, *which not only leads to a robust model for our instance reweighted optimization problem (7) but also provides a solution to the generic compositional bilevel optimization problem.*

As alluded earlier, solving the compositional bilevel optimization problem is nontrivial. More specifically, it can be shown that the gradient of the total objective is $\nabla F(\theta) = \frac{\partial g(\theta, \delta^*(\theta))}{\partial \theta} \nabla f\left(g\left(\theta, \delta^*(\theta)\right)\right)$ by applying the chain rule. Due to the fact that $\nabla f(\cdot)$ needs to be evaluated at the full value $g\left(\theta, \delta^*(\theta)\right)$, standard stochastic gradient descent methods cannot be naively applied here. The reason is that even if we can obtain the unbiased estimates $g_i\left(\theta, \delta_i^*(\theta)\right)$, the product $\frac{\partial g_i(\theta, \delta_i^*(\theta))}{\partial \theta} \nabla f\left(g_i\left(\theta, \delta_i^*(\theta)\right)\right)$ would still be biased, unless $f(\cdot)$ is a linear function. This key difference makes problem (8) particularly challenging and sets it apart from the standard finite-sum bilevel optimization problem in which the total objective is linear w.r.t. the sampling probabilities $\frac{1}{M}$.

To design a theoretically grounded algorithm for problem (8), note that the stochastic compositional gradient descent (SCGD) Wang et al. (2017) algorithm for the single-level compositional optimization problem keeps track of a running average of the composed function evaluations during the algorithm running. Inspired by SCGD, we propose a novel algorithm (see Algorithm 1) that exploits the implicit differentiation technique to deal with the bilevel aspect of problem (8). Using the implicit function theorem, we can obtain

$$\frac{\partial g_i\left(\theta, \delta_i^*(\theta)\right)}{\partial \theta} = \nabla_\theta g_i\left(\theta, \delta_i^*(\theta)\right) - \nabla_\theta \nabla_\delta h_i\left(\theta, \delta_i^*(\theta)\right) v_i^*, \tag{9}$$

with each $v_i^*$ being the solution of the linear system $\nabla_\delta^2 h_i\left(\theta, \delta_i^*(\theta)\right) v = \nabla_\delta g_i\left(\theta, \delta_i^*(\theta)\right)$.

Specifically, at each step $t$, the algorithm first samples a batch $\mathcal{B}$ of cost functions $\{(g_i, h_i)\}$ and applies $K$ steps of projected gradient descent to obtain $\delta_i^K(\theta_t)$ as an estimate of the minimizer $\delta_i^*(\theta_t)$ of each $h_i(\theta_t, \cdot)$ in $\mathcal{B}$. Then, the algorithm computes an approximation $\widehat{\nabla} g_i(\theta_t, \delta_i^K(\theta_t))$ of the stochastic gradient sample $\frac{\partial g_i(\theta_t, \delta^*(\theta_t))}{\partial \theta}$ by replacing each $\delta_i^*(\theta_t)$ with $\delta_i^K(\theta_t)$ in Equation (9). The running estimate $u_t$ of $\frac{\partial g(\theta, \delta^*(\theta))}{\partial \theta}$ and the parameters $\theta$ will be next updated as follows

$$u_{t+1} = (1 - \eta_t) u_t + \frac{\eta_t}{|\mathcal{B}|} \sum_{i=1}^{|\mathcal{B}|} g_i(\theta_t, \delta_i^K(\theta_t)) \text{ and } \theta_{t+1} = \theta_t - \frac{\beta_t}{|\mathcal{B}|} \sum_{i=1}^{|\mathcal{B}|} \widehat{\nabla} g_i(\theta_t, \delta_i^K(\theta_t)) \nabla f(u_{t+1}). \tag{10}$$

Note that we will refer the instantiation of Algorithm 1 for solving the instance reweighted problem (7) as DONE (which stands for Doubly Robust Instance Reweighted AT).

---

**Algorithm 1** Compositional Implicit Differentiation (CID)

---

1: **Input:** stepsizes $\alpha$, $\{\beta_t\}$, $\{\eta_t\}$, initializations $\theta_0 \in \mathbb{R}^d$, $\delta^0 \in \mathbb{R}^p$, and $u_0 \in \mathbb{R}^m$.
2: **for** $k = 0, 1, 2, ..., T-1$ **do**
3:     Draw a minibatch of cost functions $\mathcal{B} = \{(g_i, h_i)\}$
4:     **for** each $(g_i, h_i) \in \mathcal{B}$ (in parallel) **do**
5:         **for** $k = 1, ..., K$ **do**
6:             Update $\delta_{i,t}^k = \Pi_C \left( \delta_{i,t}^{k-1} - \alpha \nabla_\delta h_i(\theta_t, \delta_{i,t}^{k-1}) \right)$
7:         **end for**
8:         Compute sample gradient estimate $\widehat{\nabla} g_i(\theta_t, \delta_{i,t}^K)$ as in Equation (9) by replacing $\delta_i^*(\theta_t)$ with $\delta_{i,t}^K$
9:     **end for**
10:    Compute $g(\theta_t, \delta_t^K; \mathcal{B}) = \frac{1}{|\mathcal{B}|} \sum_{i=1}^{|\mathcal{B}|} g_i(\theta_t, \delta_{i,t}^K)$ and $\widehat{\nabla} g(\theta_t, \delta_t^K; \mathcal{B}) = \frac{1}{|\mathcal{B}|} \sum_{i=1}^{|\mathcal{B}|} \widehat{\nabla} g_i(\theta_t, \delta_{i,t}^K)$
11:    Update $u_{t+1} = (1 - \eta_t)u_t + \eta_t g(\theta_t, \delta_t^K; \mathcal{B})$
12:    Update $\theta_{t+1} = \theta_t - \beta_t \widehat{\nabla} g(\theta_t, \delta_t^K; \mathcal{B}) \nabla f(u_{t+1})$
13: **end for**

---

## 3.4 CONVERGENCE ANALYSIS OF CID

In the following, we establish the convergence rate of the proposed CID algorithm under widely adopted assumptions in bilevel and compositional optimization literatures (see Appendix E for the statement of assumptions and proof of Theorem 1).

**Theorem 1.** *Suppose that Assumptions 1, 2, 3 (which are given in Appendix) hold. Select the stepsizes as $\beta_t = \frac{1}{\sqrt{T}}$ and $\eta_t \in [\frac{1}{2}, 1)$, and batchsize as $\mathcal{O}(T)$. Then, the iterates $\theta_t, t = 0, ..., T-1$ of the CID algorithm satisfy*

$$\frac{\sum_{t=0}^{T-1} \mathbb{E}\left\|\nabla F(\theta_t)\right\|^2}{T} \leq \mathcal{O}\left( \frac{1}{\sqrt{T}} + (1 - \alpha\mu)^K \right),$$

The proof can be found in the Appendix. Theorem 1 indicates that Algorithm 1 can achieve an $\epsilon$-accurate stationary point by selecting $T = \mathcal{O}(\epsilon^{-2})$ and $K = \mathcal{O}(\log \frac{1}{\epsilon})$. The dependency on the batchsize can be reduced to $\mathcal{O}(\epsilon^{-1})$ by selecting $\eta_t = T^{-0.25}$, which would also lead to a higher iteration complexity of $\mathcal{O}(\epsilon^{-4})$.

## 4 EXPERIMENTS

### 4.1 EXPERIMENTAL SETUP

**Datasets and Baselines.** We consider image classification problems and compare the performance of our proposed DONE method with related baselines on four image recognition datasets CIFAR10 Krizhevsky & Hinton (2009), SVHN Netzer et al. (2011), STL10 Coates et al. (2011), and GTSRB Stallkamp et al. (2012). More details about the datasets can be found in the appendix. We compare against standard adversarial training methods AT Madry et al. (2017) and FAT Zhang et al. (2020a), and three other state-of-the-art instance re-weighted adversarial training methods GAIRAT Zhang et al. (2020b), WMMR Zeng et al. (2021), and MAIL Liu et al. (2021a). We use the official publicly available codes of the respective baselines and their recommended training configurations. For our algorithm DONE, we consider three implementations based on how we solve the inner loop optimization: (i) DONE-GD uses simple non-sign projected gradient descent steps; (ii) DONE-ADAM employs the Adam optimizer; and (iii) DONE-PGD adopts the projected gradient sign method. We run all baselines on a single NVIDIA Tesla V100 GPU.

More details about the training and hyperparameters search can be found in Appendix B.

**Evaluation.** For all baselines, we report their standard accuracy on clean data (SA), the robust accuracy against 20 steps PGD attacks (RA-PGD) Madry et al. (2017), the robust accuracy against AutoAttacks (RA-AA) Croce & Hein (2020), and the RA-PGD of the 30% most vulnerable classes (RA-Tail-30) as a measure of robustness against attacks on the most vulnerable data points.

### 4.2 BETTER DISTRIBUTION OF ROBUST PERFORMANCE

We first demonstrate that our proposed doubly robust formulation can indeed achieve robust performance in a more balanced way across the empirical distribution. Figure 1 shows the per class

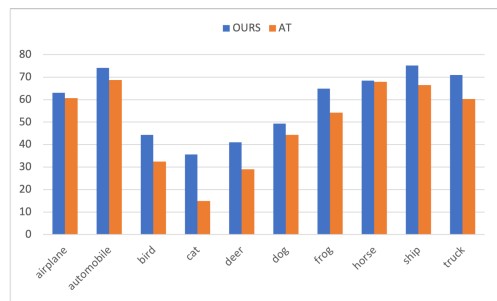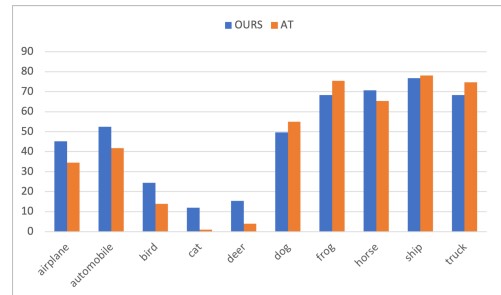

Figure 1: Per-class robust accuracy comparisons between our method and traditional AT method on balanced and imbalanced (0.2 imbalance ratio) CIFAR10.

Table 1: Performance evaluations on balanced and imbalanced (0.2 imbalance ratio) CIFAR10.

| Method | Balanced CIFAR10 | | | | Unbalanced CIFAR10 | | | |
|---|---|---|---|---|---|---|---|---|
| | SA | RA-PGD | RA-Tail-30 | RA-AA | SA | RA-PGD | RA-Tail-30 | RA-AA |
| AT | 82.1 | 49.29 | 28.35 | 45.22 | 69.74 | 42.37 | 6.25 | 39.55 |
| FAT | **86.21** | 46.59 | 27.12 | 43.71 | - | - | - | - |
| WMMR | 81.6 | 49.53 | 31.24 | 40.9 | - | - | - | - |
| MAIL | 83.47 | 55.12 | 37.30 | 44.08 | 72.01 | 45.64 | 9.8 | 37.17 |
| GAIRAT | 83.22 | 54.81 | 37.45 | 41.10 | 73.87 | 45.18 | 16.9 | 35.43 |
| DONE-GD | 83.41 | 57.46 | 40.11 | **45.66** | 74.22 | **48.29** | **17.19** | **40.06** |
| DONE-PGD | 82.62 | **58.54** | 40.18 | 44.49 | **74.58** | 48.13 | 15.83 | 38.69 |
| DONE-ADAM | 82.25 | 58.51 | **40.36** | 44.20 | 74.56 | 48.15 | 17.10 | 39.46 |

robust accuracy (RA-PGD) of the standard AT method and our doubly-robust approach (i.e., vanilla DONE-GD method) for both balanced and imbalanced (with an imbalance ratio of 0.2) CIFAR10 dataset. For the balanced case, our algorithm improves the robustness on all classes, meanwhile with a more significant boost on the weakest classes (*cat*, *deer*, and *bird*). On the other hand, for the imbalanced data case, the classes with more examples (last five categories) heavily dominate the robust training dynamic. This consequently leads to very high robustness on those classes, but nearly zero robustness on the vulnerable classes (such as *cat*). However, our method can still boost the per class RA-PGD on the weak classes (+11% on average on the 3 most vulnerable classes) and at the same time maintain a superior average RA-PGD. Overall, the results for both balanced and imbalanced settings clearly demonstrate that our doubly-robust approach can, in fact, improve worst-case robustness and hence achieve superior average robust performance.

## 4.3 MAIN RESULTS

**Comparisons under CIFAR10.** The overall performance of the compared baselines under both balanced and imbalanced CIFAR10 are reported in Table 1. We highlight the following important observations. **First,** overall our methods outperform all other baselines in terms of all three robustness metrics (RA-PGD, RA-Tail-30, and RA-AA), meanwhile also maintaining a competitive standard accuracy (SA). In particular, our algorithms can

Table 4: Comparisons with fast AT methods.

| Method | SA | RA-PGD | RA-Tail-30 |
|---|---|---|---|
| Fast-AT | **82.44** | 45.37 | 23.3 |
| Fast-AT-GA | 79.83 | 47.56 | 25.01 |
| Fast-BAT | 79.91 | 49.13 | 26.05 |
| DONE | 79.17 | **55.17** | **37.13** |

improve the RA-PGD of the strongest baseline (MAIL) by over 3% with most of the gain coming from improvement on the weakest classes, as is depicted on the RA-Tail-30 column. This shows that our doubly robust approach can mitigate the weak robustness on the vulnerable data points while also keeping the robust performance on well guarded examples (i.e., easy data points) at the same level. **Second**, note that the instance reweighted baselines consistently outperform the methods without reweighting on the RA-Tail-30 metric, which indicates that reweighting in general boosts the robustness on weak examples. This advantage is even clearer on the imbalanced data case. Yet, our algorithms still outperform the other instance reweighted methods by around 3% in terms of RA-Tail-30 in the balanced data setup due to their doubly-robust nature, which clearly is helpful both for average and worst-case robust performance. **Third**, note that the other methods that employ

Table 2: Performance evaluations on balanced and imbalanced (0.2 imbalance ratio) SVHN.

| Method | Balanced SVHN | | | | Unbalanced SVHN (0.2) | | | |
|---|---|---|---|---|---|---|---|---|
| | SA | RA-PGD | RA-Tail-30 | RA-AA | SA | RA-PGD | RA-Tail-30 | RA-AA |
| AT | **93.21** | 57.82 | 47.21 | 46.27 | 88.46 | 51.08 | 33.67 | 41.13 |
| MAIL | 93.11 | 65.56 | 52.23 | 41.38 | 86.62 | 48.48 | 31.91 | 34.46 |
| GAIRAT | 91.56 | 64.74 | 52.15 | 39.41 | 86.73 | 53.79 | 36.46 | 33.25 |
| DONE-PGD | 92.80 | **66.20** | **55.84** | 48.32 | 88.05 | 54.85 | 39.91 | 41.44 |
| DONE-ADAM | 92.58 | 65.72 | 53.79 | **49.13** | **88.98** | **55.90** | **41.10** | **42.38** |

Table 3: Performance evaluations on STL10 and GTSRB (originally imbalanced) datasets.

| Method | STL10 | | | | GTSRB | | | |
|---|---|---|---|---|---|---|---|---|
| | SA | RA-PGD | RA-Tail-30 | RA-AA | SA | RA-PGD | RA-Tail-30 | RA-AA |
| AT | 67.11 | 36.28 | 10.07 | 32.58 | 88.13 | 59.65 | 27.03 | **57.83** |
| MAIL | **68.06** | 38.20 | 13.33 | 32.86 | 88.47 | 55.96 | 20.73 | 53.44 |
| GAIRAT | 65.67 | 35.23 | 15.21 | 30.42 | 86.67 | 54.38 | 22.10 | 51.18 |
| DONE-PGD | 66.98 | **40.23** | **17.87** | 33.71 | **89.34** | **60.16** | 27.41 | 57.25 |
| DONE-ADAM | 66.92 | 39.70 | 17.62 | **34.59** | 88.76 | 60.05 | **28.35** | 57.70 |

heuristics to compute the instance weights achieve worst RA-AA performance compared to the standard AT method. In contrast, our algorithms, which also fall in the instance reweighted paradigm, can still attain competitive performance for RA-AA compared to the standard AT method. This highlights the suboptimality of using heuristics which could be geared towards improving one metric (such as the RA-PGD) but may not be necessarily beneficial to the overall robustness of the model.

**Performance Comparisons on the other datasets.** Table 2 shows the evaluations of the compared baselines on the SVHN dataset. As depicted, our algorithms (DONE-PGD and DONE-ADAM) significantly outperform the standard AT method on the RA-PGD metric and at the same time achieve better robustness against AutoAttacks (RA-AA). Compared with the instance reweighted baselines (MAIL & GAIRAT), the advantage of our methods is even more important on the RA-AA metric (e.g., up to around $+8\%$ on RA-AA vs $+1.5\%$ on RA-PGD for the balanced data setting). We also note considerable improvements on the GTSRB and STL10 datasets in Table 3. Similarly to the CIFAR10 dataset, our approach yields an important boost on the RA-Tail-30 robustness metric compared to all other baselines and the advantage is more significant on the imbalanced data case. These results consistently demonstrate that our doubly-robust approach can indeed improve worst-case robust performance meanwhile also maintaining/improving the overall robustness.

**Evaluations under Fast AT Setting.** We also compare our approach with fast adversarial training methods. For this setup, we generate the adversarial attacks during training with only 1 GD step after initialization with 1 PGD warm-up step Zhang et al. (2022) and train all baselines for 25 epochs. We compare our method with Fast-BAT Zhang et al. (2022), Fast-AT Wong et al. (2020), and Fast-AT-GA Andriushchenko & Flammarion (2020) on CIFAR10. The evaluations of the compared methods are reported in Table 4. Our algorithm achieves a much better robust performance and at the same time keeps a competitive SA. In particular, we note a significant boost $(+11\%)$ in RA-Tail-30, which is mainly the cause of the improvement in the overall RA-PGD.

## 5 CONCLUSIONS

In this paper, we proposed a novel doubly robust instance reweighted adversarial training framework based on DRO and bilevel optimization, which not only determines the instance weights for AT in a theoretically grounded way but also addresses the non-uniform issues of traditional AT by boosting the robust performance of the most vulnerable examples. To address the technical challenges in solving the doubly robust optimization problem, we proposed a penalized reformulation using the log-barrier penalty method, and developed a novel algorithm based on implicit function theorem and tracking a running average of the outer level function values. Our proposed framework also leads to a new finite-sum compositional bilevel optimization problem, which can be of great interest to the optimization community and solved by our developed algorithm with theoretical guarantee. In the experiments on standard benchmarks, our doubly-robust approach (DONE) outperforms related state-of-the-art baseline approaches in average robust performance and also improves the robustness against attacks on the weakest data points.

ACKNOWLEDGEMENTS

The work of D. Sow and Y. Liang was supported in part by the U.S. National Science Foundation under the grants CCF-1900145, ECCS-2113860 and CNS-2112471.

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

SUPPLEMENTARY MATERIAL

We provide the details omitted in the main paper. The sections are organized as fellows:

• Appendix A: We discuss related literature about bilevel optimization and stochastic compositional optimization.

• Appendix B: We provide more details about datasets, training setups, hyperparameters search, and implementations.

• Appendix C: We provide the distributions of the robustly learned instance weights and models' confusion matrices for the considered datasets.

• Appendix D: We provide the proof of Proposition 1.

• Appendix E: We present the convergence analysis of our proposed CID algorithm including the statements of assumptions and the full proof of Theorem 1.

## A    MORE RELATED WORK DISCUSSIONS ABOUT BILEVEL OPTIMIZATION AND STOCHASTIC COMPOSITIONAL OPTIMIZATION

**Bilevel optimization** Bilevel optimization is a powerful tool to study many machine learning applications such as hyperparameter optimization (Franceschi et al., 2018; Shaban et al., 2019), meta-learning (Bertinetto et al., 2018; Franceschi et al., 2018; Rajeswaran et al., 2019; Ji et al., 2020; Liu et al., 2021b), neural architecture search (Liu et al., 2018; Zhang et al., 2021a), etc. Existing approaches are usually approximate implicit differentiation (AID) based (Domke, 2012; Pedregosa, 2016; Gould et al., 2016; Liao et al., 2018; Lorraine et al., 2020), or iterative differentiation (ITD) based (Domke, 2012; Maclaurin et al., 2015; Franceschi et al., 2017; Finn et al., 2017; Shaban et al., 2019; Rajeswaran et al., 2019; Liu et al., 2020). The convergence rates of these methods have been widely established (Grazzi et al., 2020a; Ji et al., 2021; Rajeswaran et al., 2019; Ji & Liang, 2021). Bilevel optimization has been leveraged in adversarial training very recently, which provides a more generic framework by allowing independent designs of the inner and outer level objectives Zhang et al. (2022). However, none of these studies investigated bilevel optimization when the outer objective is in the form of compositions of functions. In this work, we introduce the compositional bilevel optimization problem as a novel pipeline for instance reweighted AT, and establish its first known convergence rate.

**Stochastic compositional optimization** Stochastic compositional optimization (SCO) deals with the minimization of compositions of stochastic functions. Wang et al. (2017) proposed the compositional stochastic gradient descent (SCGD) algorithm as a pioneering method for SCO problems and established its convergence rate. Many extentions of SCGD have been proposed with improved rates, including accelerated and adaptive SCGD methods Wang et al. (2016); Tutunov et al. (2020), and variance reduced SCGD methods Lian et al. (2017); Blanchet et al. (2017); Lin et al. (2020); Devraj & Chen (2019); Hu et al. (2019). A SCO reformulation has also been used to solve nonconvex distributionally robust optimization (DRO) Rahimian & Mehrotra (2019); Qian et al. (2019) problems. The problem studied in this paper naturally falls into a new class of problems but with an additional inner loop compared to the existing single-level SCO problem, which we refer to as compositional bilevel optimization (CBO).

## B    MORE EMPIRICAL SPECIFICATIONS

### B.1    MORE DETAILS ABOUT TRAINING AND HYPERPARAMETERS SEARCH

Following the standard practice in adversarial training Madry et al. (2017); Liu et al. (2021a); Zhang et al. (2020b), we train our baselines using stochastic gradient descent with a minibtach size of 128 and a momentum of 0.9. We use ResNet-18 as the backbone network as in Madry et al. (2017) and train our baselines for 60 epochs with a cyclic learning rate schedule where the maximum learning rate is set to 0.2 Zhang et al. (2020b); Liu et al. (2021c) (please see fig. 2). We consider $\ell_\infty$-norm bounded adversarial perturbations with a maximum radius of $\epsilon = 8/255$ both for training and testing.

For the KL-divergence regularization parameter $r$ in our algorithms, we use a decayed schedule where we initially set it to 10 and decay it to 1 and 0.1, respectively at epochs 40 and 50 (see fig. 2). This setting allows our methods to start with an instance-weight distribution close to uniform at the beginning of training where the weights are less informative, and progressively emphasize more on learning a weight distribution that boosts worst-case adversarial robustness. All hyperparameters were fixed by holding out 10% of the training data as a validation set and selecting the values that achieve the best performance on the validation set. For the reported results, we train on the full training dataset and report the performance on the testing set Zhang et al. (2020b); Liu et al. (2021a).

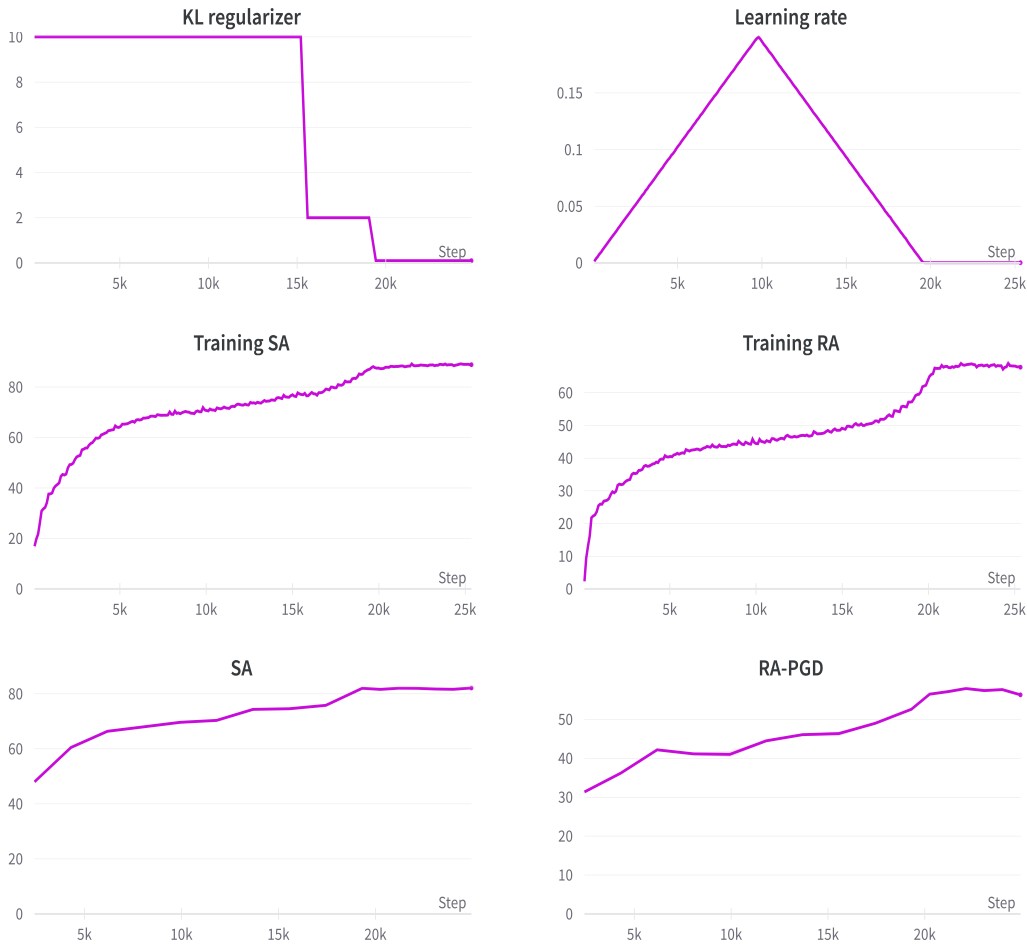

Figure 2: Learning process of our method DONE-ADAM for the balanced CIFAR10 experiment. The **SA** and **RA-PGD** in third row are evaluated on the test set. The plots are obtained by averaging three different runs.

## B.2 FURTHER DESCRIPTIONS ABOUT DATASETS

We consider image recognition problems and compare the performance of the baselines on four datasets: CIFAR10 Krizhevsky & Hinton (2009), SVHN Netzer et al. (2011), STL10 Coates et al. (2011), and GTSRB Stallkamp et al. (2012). For CIFAR10, SVHN, and STL10 we use the training and test splits provided by Torchvision. For GTSRB, we use the splits provided in Zhang et al. (2022). STL10 has 10 categories that are similar to those in CIFAR10 but with larger colour images ($96 \times 96$ resolution) and less samples (500 per class for training and 800 per class for testing). The German Traffic Sign Recognition Benchmark (GTSRB) contains 43 classes of traffic signs, split into 39,209

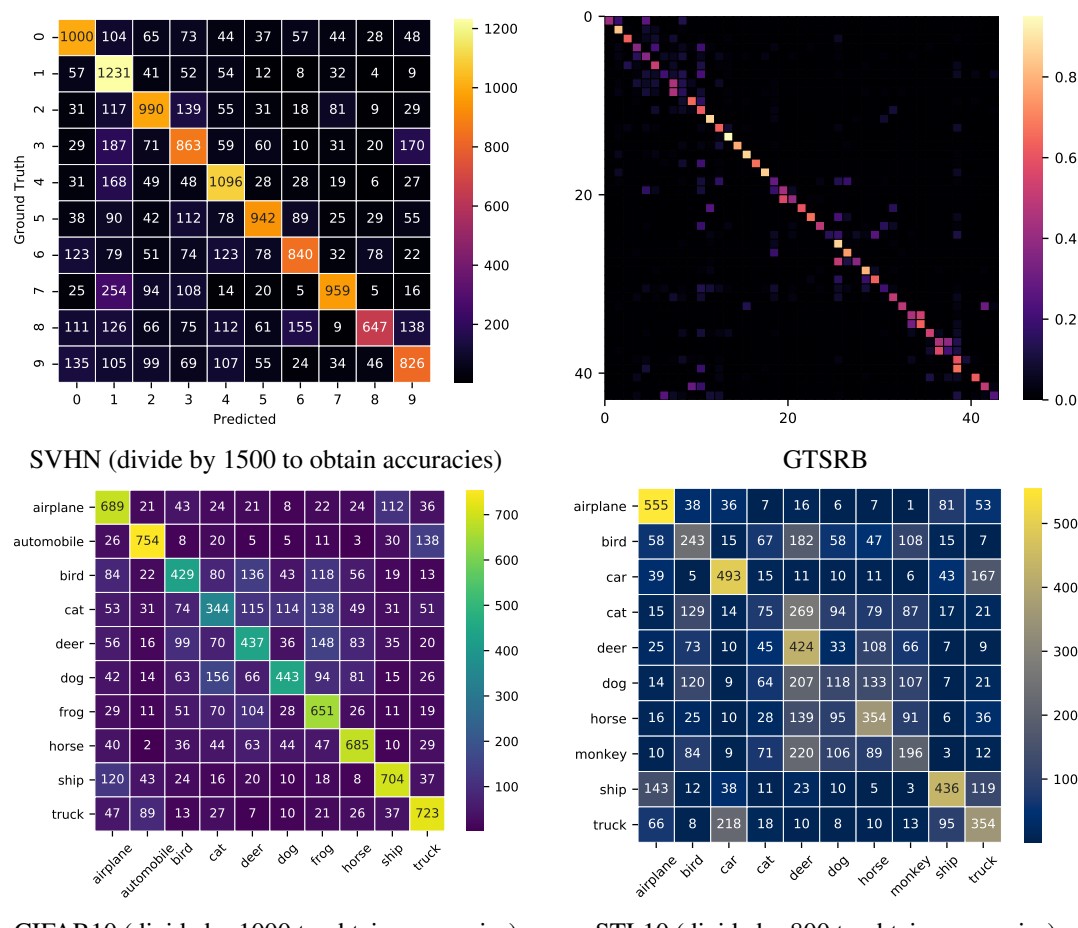

Figure 3: Confusion matrices of models robustly trained using our approach. The annotations correspond to the raw number of adversarial examples from class i that were classified as class j. Per-class robust performance are depicted in the diagonals. Axis labels are provided in first plot only.

training images and 12,630 test images. The images are $32 \times 32$ resolution colour. The dataset is highly class-imbalanced with some classes having over 2000 samples and others only 200 samples.

### B.3    FURTHER IMPLEMENTATION SPECIFICATIONS

We use the official publicly available codes of the respective baselines and their recommended training configurations. Pytorch codes for our method are provided in the supplementary material of our submission. Our implementation is built upon the codebase accompanying the paper Zhang et al. (2022). We thank the authors for making it publicly available. All codes are tested with Python 3.7 and Pytorch 1.8.

For example, to run our DONE-ADAM algorithm on the balanced CIFAR10 dataset, please run the following command:

```
python main.py --mode ours++ --dataset CIFAR10 --ir 1. --kl_coef 10
--epochs 60 --cyclic_milestone 25 --klc_milestone1 40
--klc_milestone2 50 --attack_rs_test 1 --attack_step_test 20
```

For example the argument 'ir' can be used to control the imbalance ratio. The argument 'mode' can be set to 'ours' or 'ours++', and selects among different implementations of our same approach.

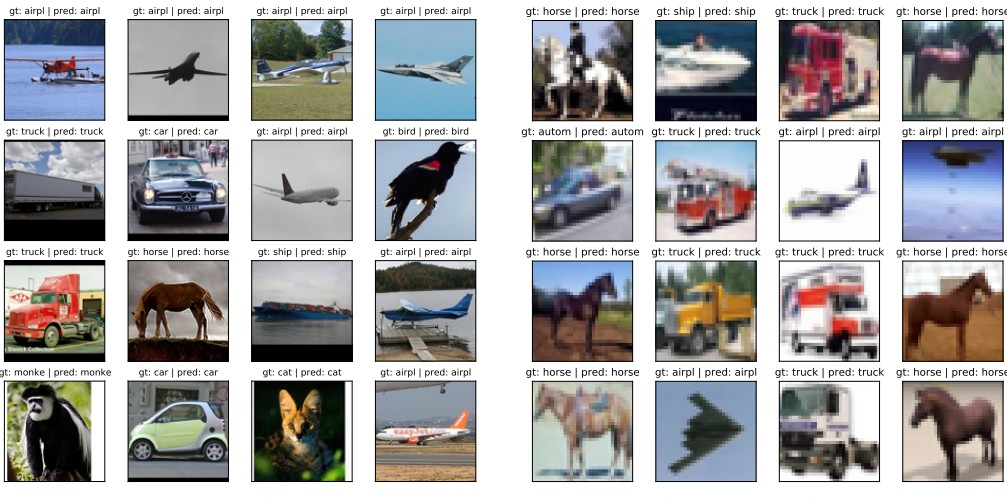

(a) STL10 dataset                  (b) CIFAR10 dataset

Figure 4: Samples with small weights from STL10 and CIFAR10 datasets. These are generally 'easy' images with the true objects well centered and clear/non-ambiguous backgrounds.

Both implementations perform similarly. Please check the file 'train.py' for more details about the arguments and the possible options.

We run all baselines on a single NVIDIA Tesla V100 GPU.

## C DISTRIBUTIONS OF LEARNED WEIGHTS

Figure 5 shows the distributions of the learned weights per-class for CIFAR10, SVHN, and STL10 datasets. The distributions are obtained on the testing sets using 20 PGD steps. Further per-class insights are also provided in Figure 3 as confusion matrices (where per-class robust accuracies are depicted in the diagonals). Comparing the two figures, we note a **negative correlation** between the magnitude of weights and the per-class robust performance, i.e., classes on which the model achieve high robustness are usually associated with weights that are closer to 0. For example, the class *automobile* in CIFAR10 datset, in which the model achieves the highest adversarial robustness of 74.5% also has around 70% of its associated weights less than 0.001. As a comparison, the most vulnerable class (i.e., *cat*, in which the model achieves a robustness of 34.4%) has more than 90% of its associated weights larger than 0.001. We note a similar correlation of the weights distributions and the robust performance in STL10 dataset. Interestingly, the robust performance is more uniformly distributed across classes in the SVHN dataset (as depicted in the corresponding confusion matrix in Figure 3) and our method was able to automatically discover very close weights distributions across classes for this dataset. This further demonstrates the generality/robustness of our approach, which can perform well no matter if instance re-weighting is advantageous or less important.

Figures 4 and 6 provides examples of images from CIFAR10 and STL10 datasets with low/high associated weights. Examples with low weights are usually 'easy' images in which the objects are well centered with clear/non-ambiguous backgrounds. Our algorithm was able to correctly classify the adversarial examples crafted from these images. In contrast, examples with high weights are generally 'hard' samples with only parts of the objects appearing or/and backgrounds that can lead to ambiguity. For example, the true label of the second image in the first row of Figure 6 is *deer* but the image also contains a car in its background, which may easily lead to confusion. Also note the first image in the third row of Figure 6, where only part of the tires of the car appears in the image.

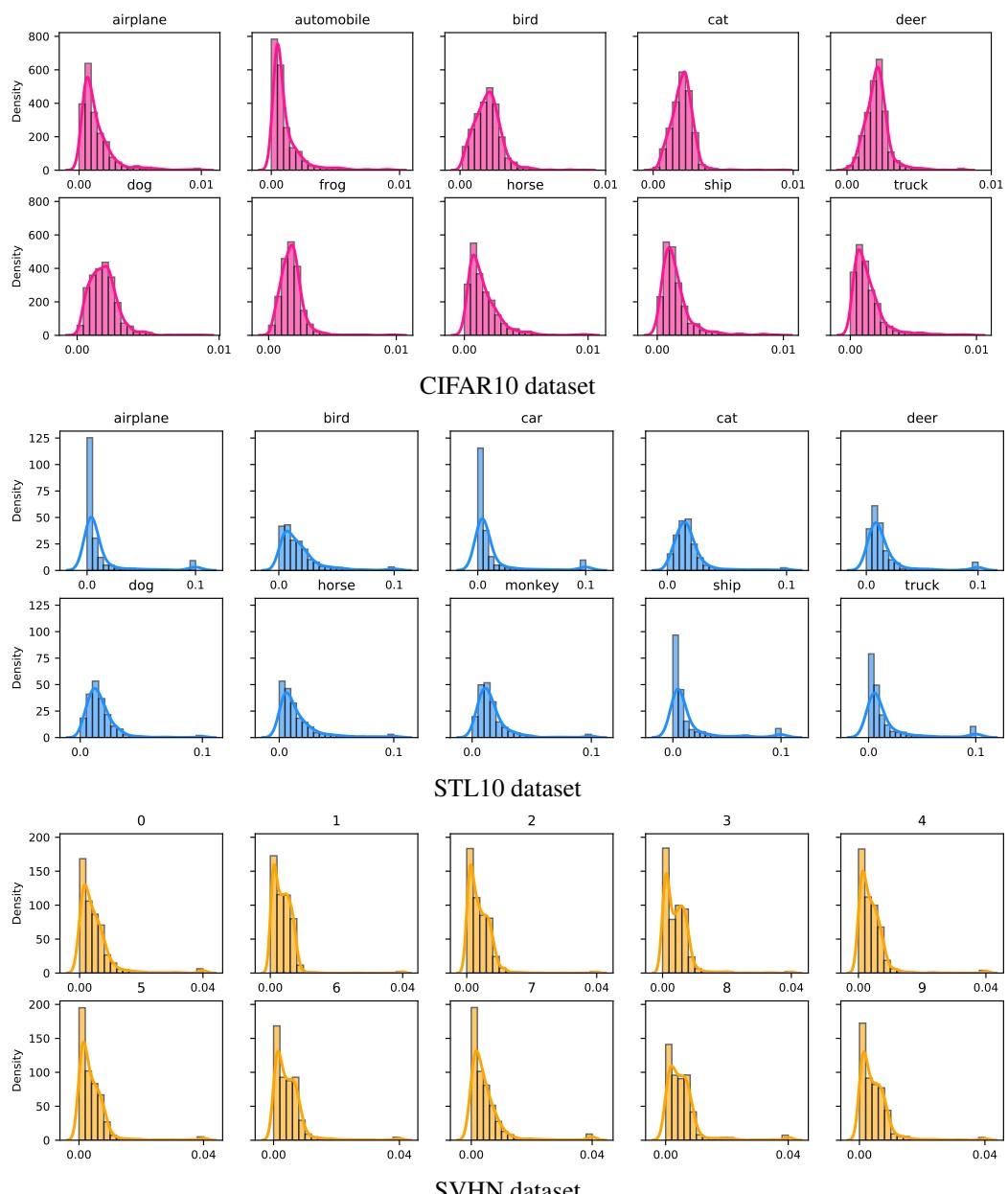

Figure 5: Distributions of the learned weights per class on the testing sets. Classes on which the model achieve high robustness are usually associated with weights that are closer to 0. For example, the class *automobile* in CIFAR10 datset, in which the model achieves the highest adversarial robustness of 74.5% also has around 70% of its associated weights less than 0.001. As a comparison, the class *cat* (in which the model achieves a robustness of 34.4%) has more than 90% of its associated weights larger than 0.001. We note a similar correlation of the weights distributions and the robust performance in STL10. The robust performance is better uniformly distributed across classes in the SVHN dataset (see fig. 3) and our method was able to obtain a similar weights distribution across classes for this dataset.

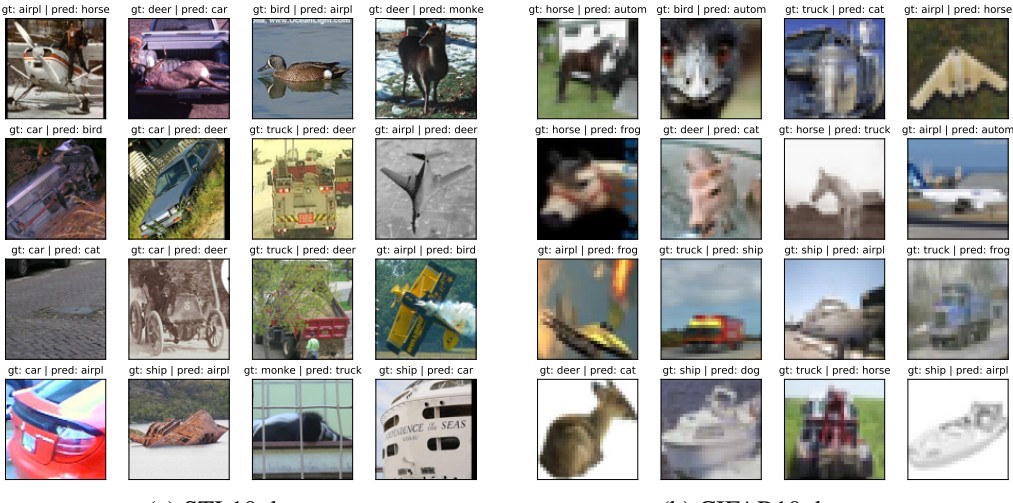

(a) STL10 dataset                    (b) CIFAR10 dataset

Figure 6: Samples with large weights from STL10 and CIFAR10 datasets. These are 'hard' examples with only parts of the objects appearing or/and complex backgrounds that easily lead to ambiguity. For example, the true label of the second image in the first row of figure (a) is *deer* but the image also contains a car in its background, which leads to ambiguity. Also note the first image in the third row of figure (a), where only part of the tires of the car appears in the image.

## D    PROOF OF PROPOSITION 1

Recall the reformulated problem (7), which we rewrite as

$$\min_{\theta} \mathcal{L}(\theta) := f\left(\frac{1}{M}\sum_{i=1}^{M} g_i(\theta, \hat{\delta}_i^*(\theta))\right)$$

$$\text{s.t. } \hat{\delta}_i^*(\theta) = \arg\min_{\delta \in \mathcal{C}_i} \ell_i^{bar}(\theta, \delta) := \ell_i'(\theta, \delta) - c\sum_{k=1}^{2p} \log(b_k - \delta^\top a_k),$$

where $g_i(\theta, \hat{\delta}_i^*(\theta)) = \exp\left(\frac{\ell_i(\theta, \hat{\delta}_i^*(\theta))}{r}\right)$, and $f(z) = r\log(z)$ for $z \geq 1$.

Applying the chain rule to the outer function, we have

$$\nabla \mathcal{L}(\theta) = \nabla f\left(\frac{1}{M}\sum_{i=1}^{M} g_i(\theta, \hat{\delta}_i^*(\theta))\right)\frac{1}{M}\sum_{i=1}^{M}\frac{\partial g_i(\theta, \hat{\delta}_i^*(\theta))}{\partial \theta}$$

$$= \frac{r}{\sum_{i=1}^{M} g_i(\theta, \hat{\delta}_i^*(\theta))}\sum_{i=1}^{M}\left(\nabla_\theta g_i(\theta, \hat{\delta}_i^*(\theta)) + \frac{\partial \hat{\delta}_i^*(\theta)}{\partial \theta}\nabla_\delta g_i(\theta, \hat{\delta}_i^*(\theta))\right). \quad (11)$$

Also, note that $\nabla_\delta \ell_i^{bar}(\theta, \delta) = \nabla_\delta \ell_i'(\theta, \delta) + c\sum_{k=1}^{2p}\frac{a_k}{b_k - \delta^\top a_k}$. Using the implicit differentiation w.r.t. $\theta$ of equation $\nabla_\delta \ell_i^{bar}(\theta, \hat{\delta}_i^*(\theta)) = \mathbf{0}$, i.e.,

$$\nabla_\delta \ell_i'(\theta, \hat{\delta}_i^*(\theta))) + c\sum_{k=1}^{2p}\frac{a_k}{b_k - a_k^\top \hat{\delta}_i^*(\theta))} = \mathbf{0},$$

we obtain

$$\nabla_{\theta\delta}\ell_i'(\theta, \hat{\delta}_i^*(\theta)) + \frac{\partial \hat{\delta}_i^*(\theta)}{\partial \theta}\nabla_\delta^2 \ell_i'(\theta, \hat{\delta}_i^*(\theta)) + c\frac{\partial \hat{\delta}_i^*(\theta)}{\partial \theta}\sum_{k=1}^{2p}\frac{a_k a_k^\top}{\left(b_k - a_k^\top \hat{\delta}_i^*(\theta)\right)^2} = \mathbf{0}.$$

Therefore, we obtain

$$\frac{\partial \hat{\delta}_i^*(\theta)}{\partial \theta} \left[ \nabla_\delta^2 \ell_i'(\theta, \hat{\delta}_i^*(\theta)) + c \sum_{k=1}^{2p} \gamma_k a_k a_k^\top \right] = -\nabla_{\theta\delta} \ell_i'(\theta, \hat{\delta}_i^*(\theta)). \tag{12}$$

where we define $\gamma_k := \frac{1}{\left(b_k - a_k^\top \hat{\delta}_i^*(\theta)\right)^2}$.

Further, note that $A = \left(I_p, -I_p\right)^\top$. Thus the first $p$ rows of $A$ (i.e., $a_k, k = 1, ..., p$) correspond to the $p$ basis vectors of $\mathbb{R}^p$, and hence $a_k a_k^\top = \mathrm{diag}(e_k)$, where $e_k$ is the $k$-th basis vector of $\mathbb{R}^p$. Thus, considering the first $p$ rows we obtain $\sum_{k=1}^p \gamma_k a_k a_k^\top = \mathrm{diag}(\gamma_1, ..., \gamma_p)$. Similarly, the bottom $p$ rows yields $\sum_{k=p+1}^{2p} \gamma_k a_k a_k^\top = \mathrm{diag}(\gamma_{p+1}, ..., \gamma_{2p})$. Therefore, we have

$$c \sum_{k=1}^{2p} \gamma_k a_k a_k^\top = c \, \mathrm{diag}(\gamma_1 + \gamma_{p+1}, ..., \gamma_p + \gamma_{2p}) := C_i(\theta). \tag{13}$$

Substituting eq. (13) in eq. (12) yields

$$\frac{\partial \hat{\delta}_i^*(\theta)}{\partial \theta} \left[ \nabla_\delta^2 \ell_i'(\theta, \hat{\delta}_i^*(\theta)) + C_i(\theta) \right] = -\nabla_{\theta\delta} \ell_i'(\theta, \hat{\delta}_i^*(\theta)).$$

If $\nabla_\delta^2 \ell_i'(\theta, \hat{\delta}_i^*(\theta)) + C_i(\theta)$ is invertable, the above equation further implies

$$\frac{\partial \hat{\delta}_i^*(\theta)}{\partial \theta} = -\nabla_{\theta\delta} \ell_i'(\theta, \hat{\delta}_i^*(\theta)) \left[ \nabla_\delta^2 \ell_i'(\theta, \hat{\delta}_i^*(\theta)) + C_i(\theta) \right]^{-1}. \tag{14}$$

Now, combining eq. (14) and eq. (11) we obtain

$$\nabla \mathcal{L}(\theta) = \frac{r}{\sum_{i=1}^M g_i(\theta, \hat{\delta}_i^*(\theta))} \sum_{i=1}^M \Big( \nabla_\theta g_i(\theta, \hat{\delta}_i^*(\theta))$$
$$- \nabla_{\theta\delta} \ell_i'(\theta, \hat{\delta}_i^*(\theta)) \left[ \nabla_\delta^2 \ell_i'(\theta, \hat{\delta}_i^*(\theta)) + C_i(\theta) \right]^{-1} \nabla_\delta g_i(\theta, \hat{\delta}_i^*(\theta)) \Big),$$

which completes the proof.

## E  CONVERGENCE ANALYSIS OF THE CID ALGORITHM

We provide the convergence analysis of the CID algorithm for solving the generic compositional bilevel optimization problem (8), which we rewrite as follows:

$$\min_\theta F(\theta) := f\left(g\left(\theta, \delta^*(\theta)\right)\right) = f\left(\frac{1}{M} \sum_{i=1}^M g_i\left(\theta, \delta_i^*(\theta)\right)\right) \tag{15}$$

$$\text{s.t. } \delta^*(\theta) = (\delta_1^*(\theta), ..., \delta_M^*(\theta)) = \mathop{\arg\min}_{(\delta_1, ..., \delta_M) \in \mathcal{V}_1 \times ... \times \mathcal{V}_M} \frac{1}{M} \sum_{i=1}^M h_i\left(\theta, \delta_i\right).$$

**Challenge and Novelty.** We note that although bilevel optimization and compositional optimization have been well studied in the optimization literature, to our best knowledge, there have not been any theoretical analysis of compositional bilevel optimization. The special challenge arising in such a problem is due to the fact that the bias error caused by the stochastic estimation of the compositional function in the outer-loop is further complicated by the approximation error from the inner loop. Our main novel development here lies in tracking such an error in the convergence analysis.

To proceed the analysis, we let $w = (\theta, \delta)$ denote all optimization parameters. We denote by $\|\cdot\|$ the $\ell_2$ norm for vectors and the spectral norm for matrices.

We adopt the following assumptions for our analysis, which are widely used in bilevel and compositional optimization literature (Grazzi et al., 2020a; Ji et al., 2021; Ji & Liang, 2021; Wang et al., 2017; Chen et al., 2021b).

**Assumption 1.** *The objective functions $f$, $g_i$, and $h_i$ for any $i = 1, \ldots, M$ satisfy*

- $f$ is $C_f$-Lipschitz continuous and $L_f$-smooth, i.e., for any $z$ and $z'$,

$$\left|f(z) - f(z')\right| \le C_f\left\|z - z'\right\|, \quad \left\|\nabla f(z) - \nabla f(z')\right\| \le L_f\left\|z - z'\right\|. \tag{16}$$

- $g_i$ is $C_g$-Lipschitz continuous and $L_g$-smooth, i.e., for any $w$ and $w'$,

$$\left\|g_i(w) - g_i(w')\right\| \le C_g\left\|w - w'\right\|, \quad \left\|\nabla g_i(w) - \nabla g_i(w')\right\| \le L_g\left\|w - w'\right\|. \tag{17}$$

- $h_i$ is $L_h$-smooth, i.e., for any $w$ and $w'$,

$$\left\|\nabla h_i(w) - \nabla h_i(w')\right\| \le L_h\left\|w - w'\right\|. \tag{18}$$

**Assumption 2.** *The function $h_i(\theta, \delta)$ for any $i = 1, \dots, M$ is $\mu$-strongly convex w.r.t. $\delta$ and its second-order derivatives $\nabla_\theta\nabla_\delta h_i(w)$ and $\nabla_\delta^2 h_i(w)$ are $L_{\theta\delta}$- and $L_{\delta\delta}$-Lipschitz, i.e., for any $w$ and $w'$,*

$$\left\|\nabla_\theta\nabla_\delta h_i(w) - \nabla_\theta\nabla_\delta h_i(w)\right\| \le L_{\theta\delta}\left\|w - w'\right\|, \quad \left\|\nabla_\delta^2 h_i(w) - \nabla_\delta^2 h_i(w')\right\| \le L_{\delta\delta}\left\|w - w'\right\|. \tag{19}$$

**Assumption 3.** *The stochastic sample $g_i$ for any $i = 1, \dots, M$ has bounded variance, i.e.,*

$$\mathbb{E}_i\left\|g_i(\theta, \delta_i) - \frac{1}{M}\sum_{j=1}^{M}g_j(\theta, \delta_j)\right\|^2 \le \sigma_g^2. \tag{20}$$

The following theorem (as restatement of Theorem 1) characterizes the convergence rate of our designed CID algorithm.

**Theorem 2** (Re-statement of Theorem 1). *Suppose that Assumptions 1, 2, 3 hold. Select the stepsizes as $\beta_t = \frac{1}{\sqrt{T}}$ and $\eta_t \in [\frac{1}{2}, 1)$, and batchsize as $|\mathcal{B}| = \mathcal{O}(T)$. Then, the iterates $\theta_t, t = 0, \dots, T - 1$ of the CID algorithm satisfy*

$$\frac{\sum_{t=0}^{T-1}\mathbb{E}\left\|\nabla F(\theta_t)\right\|^2}{T} \le \mathcal{O}\left(\frac{1}{\sqrt{T}} + (1 - \alpha\mu)^K\right)$$

In the following two subsections, we first establish a number of useful supporting lemmas and then provide the proof of Theorem 2 (which is a restatement of Theorem 1).

### E.1 SUPPORTING LEMMAS

For notational convenience, we let $L = \max\{L_f, L_g, L_h\}$, $C = \max\{C_f, C_g\}$, and $\tau = \max\{L_{\theta\delta}, L_{\delta\delta}\}$.

**Lemma 1.** *Suppose that Assumptions 1 and 2 hold. Then, the total objective $F(\theta)$ (defined at the outer level of problem (15) is $L_F$-smooth, i.e., for any $\theta$, $\theta'$,*

$$\left\|\nabla F(\theta) - \nabla F(\theta')\right\| \le L_F\left\|\theta - \theta'\right\|, \tag{21}$$

*where $L_F = C^2L\left(1 + \frac{L}{\mu}\right)^2 + CL_G$.*

*Proof.* Applying the chain rule, we have

$$\nabla F(\theta) = \frac{\partial g\left(\theta, \delta^*(\theta)\right)}{\partial \theta}\nabla f\left(g\left(\theta, \delta^*(\theta)\right)\right). \tag{22}$$

Therefore, using triangle inequality, we obtain

$$\begin{aligned}
\left\|\nabla F(\theta) - \nabla F(\theta')\right\| =& \left\|\frac{\partial g\left(\theta, \delta^*(\theta)\right)}{\partial \theta}\nabla f\left(g\left(\theta, \delta^*(\theta)\right)\right) - \frac{\partial g\left(\theta', \delta^*(\theta')\right)}{\partial \theta}\nabla f\left(g\left(\theta', \delta^*(\theta')\right)\right)\right\| \\
\le& \left\|\frac{\partial g\left(\theta, \delta^*(\theta)\right)}{\partial \theta}\left(\nabla f\left(g\left(\theta, \delta^*(\theta)\right)\right) - \nabla f\left(g\left(\theta', \delta^*(\theta')\right)\right)\right)\right\| \\
& + \left\|\left(\frac{\partial g\left(\theta, \delta^*(\theta)\right)}{\partial \theta} - \frac{\partial g\left(\theta', \delta^*(\theta')\right)}{\partial \theta}\right)\nabla f\left(g\left(\theta', \delta^*(\theta')\right)\right)\right\|
\end{aligned}$$

$$
\leq \Big\| \frac{\partial g\left(\theta, \delta^*(\theta)\right)}{\partial \theta} \Big\| \Big\| \nabla f\left(g\left(\theta, \delta^*(\theta)\right)\right) - \nabla f\left(g\left(\theta', \delta^*(\theta')\right)\right) \Big\|
$$
$$
+ \Big\| \frac{\partial g\left(\theta, \delta^*(\theta)\right)}{\partial \theta} - \frac{\partial g\left(\theta', \delta^*(\theta')\right)}{\partial \theta} \Big\| \Big\| \nabla f\left(g\left(\theta', \delta^*(\theta')\right)\right) \Big\|
$$
$$
\leq L_f \Big\| \frac{\partial g\left(\theta, \delta^*(\theta)\right)}{\partial \theta} \Big\| \Big\| g\left(\theta, \delta^*(\theta)\right) - g\left(\theta', \delta^*(\theta')\right) \Big\|
$$
$$
+ C_f \Big\| \frac{\partial g\left(\theta, \delta^*(\theta)\right)}{\partial \theta} - \frac{\partial g\left(\theta', \delta^*(\theta')\right)}{\partial \theta} \Big\|. \tag{23}
$$

The chain rule yields

$$
\frac{\partial g_i\left(\theta, \delta_i^*(\theta)\right)}{\partial \theta} = \nabla_\theta g_i\left(\theta, \delta_i^*(\theta)\right) + \frac{\partial \delta_i^*(\theta)}{\partial \theta} \nabla_\delta g_i\left(\theta, \delta_i^*(\theta)\right)
$$
$$
= \nabla_\theta g_i\left(\theta, \delta_i^*(\theta)\right) - \nabla_\theta \nabla_\delta h_i\left(\theta, \delta_i^*(\theta)\right) \left[ \nabla_\delta^2 h_i\left(\theta, \delta_i^*(\theta)\right) \right]^{-1} \nabla_\delta g_i\left(\theta, \delta_i^*(\theta)\right),
$$

where the last equality follows from the implicit differentiation result for bilevel optimization Pedregosa (2016); Ji et al. (2021).

Thus, we obtain

$$
\Big\| \frac{\partial g_i\left(\theta, \delta_i^*(\theta)\right)}{\partial \theta} \Big\|
$$
$$
\leq \Big\| \nabla_\theta g_i\left(\theta, \delta_i^*(\theta)\right) \Big\| + \Big\| \nabla_\theta \nabla_\delta h_i\left(\theta, \delta_i^*(\theta)\right) \left[ \nabla_\delta^2 h_i\left(\theta, \delta_i^*(\theta)\right) \right]^{-1} \nabla_\delta g_i\left(\theta, \delta_i^*(\theta)\right) \Big\|
$$
$$
\leq C_g + \frac{L}{\mu} C_g. \tag{24}
$$

Therfore, $g\left(\theta, \delta^*(\theta)\right) = \frac{1}{M} \sum_{i=1}^M g_i\left(\theta, \delta_i^*(\theta)\right)$ is Lipschitz with constant $C_G = C_g\left(1 + \frac{L}{\mu}\right)$.

Further, following from Lemma 2 in Ji et al. (2021) we obtain that $\frac{\partial g(\theta, \delta^*(\theta))}{\partial \theta}$ is Lipschitz with the constant $L_G$. Thus, combining with eq. (23), we obtain

$$
\left\| \nabla F(\theta) - \nabla F(\theta') \right\| \leq L_f C_G^2 \left\| \theta - \theta' \right\| + C_f L_G \left\| \theta - \theta' \right\| \tag{25}
$$

Rearranging the above equation completes the proof. $\qquad \square$

**Lemma 2.** *Suppose that Assumptions 1 and 3 hold. Then, we have*

$$
\mathbb{E}_{\mathcal{B}} \left\| u_{t+1} - g(\theta_t, \delta_t^K) \right\|^2 \leq (1 - \eta_t) \left\| u_t - g(\theta_{t-1}, \delta_{t-1}^K) \right\|^2 + \frac{2\eta_t^2}{|\mathcal{B}|} \sigma_g^2 + \frac{C^2}{\eta_t}(1 + \kappa^2) \left\| \theta_t - \theta_{t-1} \right\|^2. \tag{26}
$$

*Proof.* We first show that $\frac{\partial \delta_{i,t}^K}{\partial \theta}$ is $\kappa$-Lipschitz. To explicitly write the dependency of $\delta_{i,t}^k$ on $\theta_t$, we define $\delta_i^k(\theta_t) := \delta_{i,t}^k$. Then we have

$$
\left\| \delta_i^K(\theta) - \delta_i^K(\theta') \right\|
$$
$$
= \left\| \Pi_{\mathcal{X}}\left( \delta_i^{K-1}(\theta) - \alpha \nabla_\delta h_i\left(\theta, \delta_i^{K-1}(\theta)\right) \right) - \Pi_{\mathcal{X}}\left( \delta_i^{K-1}(\theta') - \alpha \nabla_\delta h_i\left(\theta', \delta_i^{K-1}(\theta')\right) \right) \right\|
$$
$$
\leq \left\| \delta_i^{K-1}(\theta) - \alpha \nabla_\delta h_i\left(\theta, \delta_i^{K-1}(\theta)\right) - \delta_i^{K-1}(\theta') + \alpha \nabla_\delta h_i\left(\theta', \delta_i^{K-1}(\theta')\right) \right\|
$$
$$
\leq \underbrace{\left\| \delta_i^{K-1}(\theta) - \delta_i^{K-1}(\theta') + \alpha\left( \nabla_\delta h_i\left(\theta', \delta_i^{K-1}(\theta')\right) - \nabla_\delta h_i\left(\theta', \delta_i^{K-1}(\theta)\right) \right) \right\|}_{T_1}
$$
$$
+ \alpha \left\| \nabla_\delta h_i\left(\theta', \delta_i^{K-1}(\theta)\right) - \nabla_\delta h_i\left(\theta, \delta_i^{K-1}(\theta)\right) \right\|
$$
$$
\leq \left( \frac{L - \mu}{L + \mu} \right) \left\| \delta_i^{K-1}(\theta) - \delta_i^{K-1}(\theta') \right\| + \alpha L \left\| \theta - \theta' \right\|,
$$

where we upper-bound the term $T_1$ using the fact that the operator $y \to y - \alpha \nabla h(y)$ is a contraction mapping with the constant $\frac{L-\mu}{L+\mu}$ for an $L$-smooth and $\mu$-strongly convex function $h$ when the stepsize $\alpha$ is set to $\frac{2}{L+\mu}$. Hence, telescoping the previous inequality over $k$ from $K-1$ down to $0$ yields

$$
\left\| \delta_i^K(\theta) - \delta_i^K(\theta') \right\| \leq \left( \frac{L-\mu}{L+\mu} \right)^K \left\| \delta_i^0(\theta) - \delta_i^0(\theta') \right\| + \alpha L \left\| \theta - \theta' \right\| \sum_{k=0}^{K-1} \left( \frac{L-\mu}{L+\mu} \right)^k
$$

$$\leq 0 + \frac{\alpha L}{1 - \frac{L-\mu}{L+\mu}}\|\theta - \theta'\| = \kappa\|\theta - \theta'\|, \tag{27}$$

where the second inequality follows because $\delta_i^0(\theta) = \delta_i^0(\theta')$ as the same initial point, and the last equality follows by setting the stepsize $\alpha$ to $\frac{2}{L+\mu}$.

Denote $d_t = (1 - \eta_t)\left(g(\theta_t, \delta_t^K) - g(\theta_{t-1}, \delta_{t-1}^K)\right) = \frac{1-\eta_t}{M}\sum_{i=1}^{M}\left(g_i(\theta_t, \delta_{i,t}^K) - g_i(\theta_{t-1}, \delta_{i,t-1}^K)\right)$. We can then obtain

$$
\begin{aligned}
\|d_t\|^2 &\leq \frac{(1-\eta_t)^2}{M}\sum_{i=1}^{M}\left\|g_i(\theta_t, \delta_{i,t}^K) - g_i(\theta_{t-1}, \delta_{i,t-1}^K)\right\|^2 \\
&\leq \frac{(1-\eta_t)^2}{M}\sum_{i=1}^{M}C^2\left(\|\theta_t - \theta_{t-1}\|^2 + \|\delta_{i,t}^K - \delta_{i,t-1}^K\|^2\right) \\
&\leq \frac{(1-\eta_t)^2}{M}\sum_{i=1}^{M}C^2(1+\kappa^2)\|\theta_t - \theta_{t-1}\|^2 \\
&= (1-\eta_t)^2(1+\kappa^2)C^2\|\theta_t - \theta_{t-1}\|^2.
\end{aligned}
\tag{28}
$$

Recall $u_{t+1} = (1 - \eta_t)u_t + \eta_t g(\theta_t, \delta_t^K; \mathcal{B})$. Thus combining with the definition of $d_t$, we have

$$
\begin{aligned}
\mathbb{E}_{\mathcal{B}}&\|u_{t+1} - g(\theta_t, \delta_t^K) + d_t\|^2 \\
&= \mathbb{E}_{\mathcal{B}}\|(1 - \eta_t)\left(u_t - g(\theta_{t-1}, \delta_{t-1}^K)\right) + \eta_t\left(g(\theta_t, \delta_t^K; \mathcal{B}) - g(\theta_t, \delta_t^K)\right)\|^2 \\
&= (1 - \eta_t)^2\|u_t - g(\theta_{t-1}, \delta_{t-1}^K)\|^2 + \eta_t^2\mathbb{E}_{\mathcal{B}}\|g(\theta_t, \delta_t^K; \mathcal{B}) - g(\theta_t, \delta_t^K)\|^2 \\
&\quad + 2(1 - \eta_t)\eta_t\left\langle u_t - g(\theta_{t-1}, \delta_{t-1}^K), \mathbb{E}_{\mathcal{B}}\left(g(\theta_t, \delta_t^K; \mathcal{B}) - g(\theta_t, \delta_t^K)\right)\right\rangle \\
&= (1 - \eta_t)^2\|u_t - g(\theta_{t-1}, \delta_{t-1}^K)\|^2 + \frac{\eta_t^2}{|\mathcal{B}|}\mathbb{E}_i\|g_i(\theta_t, \delta_{i,t}^K) - g(\theta_t, \delta_t^K)\|^2 \\
&\leq (1 - \eta_t)^2\|u_t - g(\theta_{t-1}, \delta_{t-1}^K)\|^2 + \frac{\eta_t^2}{|\mathcal{B}|}\sigma_g^2.
\end{aligned}
\tag{29}
$$

Based on the inequality $\|a + b\|^2 \leq (1 + c)\|a\|^2 + (1 + \frac{1}{c})\|b\|^2$ for any $c > 0$, by letting $c = \eta_t$, we have

$$
\begin{aligned}
\mathbb{E}_{\mathcal{B}}\|u_{t+1} - g(\theta_t, \delta_t^K)\|^2 &\leq (1 + \eta_t)\mathbb{E}_{\mathcal{B}}\|u_{t+1} - g(\theta_t, \delta_t^K) + d_t\|^2 + (1 + \frac{1}{\eta_t})\mathbb{E}_{\mathcal{B}}\|d_t\|^2 \\
&\leq (1 + \eta_t)(1 - \eta_t)^2\|u_t - g(\theta_{t-1}, \delta_{t-1}^K)\|^2 + \frac{(1 + \eta_t)\eta_t^2}{|\mathcal{B}|}\sigma_g^2 \\
&\quad + \frac{1 + \eta_t}{\eta_t}(1 - \eta_t)^2(1 + \kappa^2)C^2\|\theta_t - \theta_{t-1}\|^2 \\
&\leq (1 - \eta_t)\|u_t - g(\theta_{t-1}, \delta_{t-1}^K)\|^2 + \frac{2\eta_t^2}{|\mathcal{B}|}\sigma_g^2 + \frac{C^2}{\eta_t}(1 + \kappa^2)\|\theta_t - \theta_{t-1}\|^2.
\end{aligned}
\tag{30}
$$

Hence, the proof is complete. □

**Lemma 3.** *Suppose that Assumptions 1 and 2 hold. Then we have*

$$\left\|\frac{\partial g\left(\theta_t, \delta^*(\theta_t)\right)}{\partial\theta} - \widehat{\nabla}g(\theta_t, \delta_t^K)\right\|^2 \leq \Omega(1 - \alpha\mu)^K\Delta_0, \tag{31}$$

*where $\Delta_0 = \max_{i,t}\|\delta_i^*(\theta_t) - \delta_0\|^2$ and $\Omega = \mathcal{O}\left(L + \frac{\tau^2 C^2}{\mu^2} + L\left(\kappa + \frac{\tau C}{\mu^2}\right)^2\right)$.*

*Proof.* The proof follows the steps similar to those in the proof of Lemma 3 in Ji et al. (2021). □

In the following, we define $\Lambda = \Omega(1 - \alpha\mu)^K\Delta_0$.

**Lemma 4.** *Suppose that Assumptions 1, 2, 3 hold. Then, we have*

$$\mathbb{E}_{\mathcal{B}}F(\theta_{t+1}) - F(\theta_t) \leq - \beta_t\alpha_t\|\nabla F(\theta_t)\|^2 + \beta_t\Gamma\Delta_0(1-\alpha\mu)^K$$

$$+ \eta_t\mathbb{E}_{\mathcal{B}}\|g(\theta_t,\delta_t^K) - u_{t+1}\|^2 + \frac{L_F\beta_t^2}{2}C^4\left(1+\frac{L}{\mu}\right)^2, \qquad (32)$$

*where $\alpha_t = \frac{1}{2} - \frac{\beta_t L^2}{\eta_t}C^2\left(1+\frac{L}{\mu}\right)^2$.*

*Proof.* Based on the Lipschitzness of $\nabla F(\theta)$ in Lemma 1, we have

$$F(\theta_{t+1}) - F(\theta_t) \leq \langle \nabla F(\theta_t), \theta_{t+1} - \theta_t \rangle + \frac{L_F}{2}\|\theta_{t+1}-\theta_t\|^2$$

$$\leq - \beta_t\|\nabla F(\theta_t)\|^2 + \beta_t\left\langle \nabla F(\theta_t), \nabla F(\theta_t) - \widehat{\nabla}g(\theta_t,\delta_t^K;\mathcal{B})\nabla f(u_{t+1}) \right\rangle$$

$$+ \frac{L_F\beta_t^2}{2}\|\widehat{\nabla}g(\theta_t,\delta_t^K;\mathcal{B})\nabla f(u_{t+1})\|^2$$

$$\leq - \beta_t\|\nabla F(\theta_t)\|^2 + \underbrace{\beta_t\left\langle \nabla F(\theta_t), \nabla F(\theta_t) - \widehat{\nabla}g(\theta_t,\delta_t^K)\nabla f\left(g(\theta_t,\delta_t^K)\right) \right\rangle}_{A_1}$$

$$+ \underbrace{\beta_t\left\langle \nabla F(\theta_t), \widehat{\nabla}g(\theta_t,\delta_t^K)\nabla f\left(g(\theta_t,\delta_t^K)\right) - \widehat{\nabla}g(\theta_t,\delta_t^K;\mathcal{B})\nabla f(u_{t+1}) \right\rangle}_{A_2}$$

$$+ \frac{L_F\beta_t^2}{2}\|\widehat{\nabla}g(\theta_t,\delta_t^K;\mathcal{B})\nabla f(u_{t+1})\|^2. \qquad (33)$$

Next, we upper-bound the inner product terms $A_1$ and $A_2$, respectively. Using Young's inequality, we obtain

$$A_1 \leq \frac{\beta_t}{2}\|\nabla F(\theta_t)\|^2 + \frac{\beta_t}{2}\|\nabla F(\theta_t) - \widehat{\nabla}g(\theta_t,\delta_t^K)\nabla f\left(g(\theta_t,\delta_t^K)\right)\|^2$$

$$\leq \frac{\beta_t}{2}\|\nabla F(\theta_t)\|^2 + \beta_t\|\frac{\partial g\left(\theta_t,\delta^*(\theta_t)\right)}{\partial\theta}\|^2\|\nabla f\left(g\left(\theta_t,\delta^*(\theta_t)\right)\right) - \nabla f\left(g(\theta_t,\delta_t^K)\right)\|^2$$

$$+ \beta_t\|\nabla f\left(g(\theta_t,\delta_t^K)\right)\|^2\|\frac{\partial g\left(\theta_t,\delta^*(\theta_t)\right)}{\partial\theta} - \widehat{\nabla}g(\theta_t,\delta_t^K)\|^2$$

$$\leq \frac{\beta_t}{2}\|\nabla F(\theta_t)\|^2 + \beta_t L_G^2 L^2\|g\left(\theta_t,\delta^*(\theta_t)\right) - g(\theta_t,\delta_t^K)\|^2$$

$$+ \beta_t C^2\|\frac{\partial g\left(\theta_t,\delta^*(\theta_t)\right)}{\partial\theta} - \widehat{\nabla}g(\theta_t,\delta_t^K)\|^2$$

$$\leq \frac{\beta_t}{2}\|\nabla F(\theta_t)\|^2 + \frac{\beta_t L_G^2 L^2}{M}\sum_{i=1}^{M}\|g_i\left(\theta_t,\delta_i^*(\theta_t)\right) - g_i(\theta_t,\delta_{i,t}^K)\|^2 + \beta_t C^2\Lambda$$

$$\leq \frac{\beta_t}{2}\|\nabla F(\theta_t)\|^2 + \frac{\beta_t L_G^2 L^2 C^2}{M}\sum_{i=1}^{M}\|\delta_i^*(\theta_t) - \delta_{i,t}^K\|^2 + \beta_t C^2\Lambda$$

$$\leq \frac{\beta_t}{2}\|\nabla F(\theta_t)\|^2 + \beta_t L_G^2 L^2 C^2\frac{(1-\alpha\mu)^K}{M}\sum_{i=1}^{M}\|\delta_i^*(\theta_t) - \delta_0\|^2 + \beta_t C^2\Lambda$$

$$\leq \frac{\beta_t}{2}\|\nabla F(\theta_t)\|^2 + \beta_t L_G^2 L^2 C^2\Delta_0(1-\alpha\mu)^K + \beta_t C^2\Omega(1-\alpha\mu)^K\Delta_0$$

$$= \frac{\beta_t}{2}\|\nabla F(\theta_t)\|^2 + \beta_t\Gamma\Delta_0(1-\alpha\mu)^K, \qquad (34)$$

where $\Gamma = L_G^2 L^2 C^2 + C^2\Omega$, $\Delta_0 = \max_{i,t}\|\delta_i^*(\theta_t) - \delta_0\|^2$, and $\Lambda = \Omega(1-\alpha\mu)^K\Delta_0$.

Further, we have

$$\mathbb{E}_{\mathcal{B}}A_2 = \beta_t\mathbb{E}_{\mathcal{B}}\left\langle \nabla F(\theta_t), \widehat{\nabla}g(\theta_t,\delta_t^K;\mathcal{B})\nabla f\left(g(\theta_t,\delta_t^K)\right) - \widehat{\nabla}g(\theta_t,\delta_t^K;\mathcal{B})\nabla f(u_{t+1}) \right\rangle$$

$$\leq \beta_t \big\| \nabla F(\theta_t) \big\| \mathbb{E}_{\mathcal{B}} \left[ \big\| \widehat{\nabla} g(\theta_t, \delta_t^K; \mathcal{B}) \big\| \big\| \nabla f \left( g(\theta_t, \delta_t^K) \right) - \nabla f(u_{t+1}) \big\| \right]$$

$$\leq \beta_t L \big\| \nabla F(\theta_t) \big\| \mathbb{E}_{\mathcal{B}} \left[ \big\| \widehat{\nabla} g(\theta_t, \delta_t^K; \mathcal{B}) \big\| \big\| g(\theta_t, \delta_t^K) - u_{t+1} \big\| \right]$$

$$\leq \eta_t \mathbb{E}_{\mathcal{B}} \big\| g(\theta_t, \delta_t^K) - u_{t+1} \big\|^2 + \frac{\beta_t^2 L^2}{\eta_t} \big\| \nabla F(\theta_t) \big\|^2 \mathbb{E}_{\mathcal{B}} \big\| \widehat{\nabla} g(\theta_t, \delta_t^K; \mathcal{B}) \big\|^2$$

$$\leq \eta_t \mathbb{E}_{\mathcal{B}} \big\| g(\theta_t, \delta_t^K) - u_{t+1} \big\|^2 + \frac{\beta_t^2 L^2}{\eta_t} C^2 (1 + \frac{L}{\mu})^2 \big\| \nabla F(\theta_t) \big\|^2, \tag{35}$$

where the last inequality uses the upper-bound $\big\| \widehat{\nabla} g_i(\theta_t, \delta_{i,t}^K) \big\| \leq C + \frac{L}{\mu} C$, which can be obtained similarly to eq. (24).

Therefore, taking the conditional expectation $\mathbb{E}_{\mathcal{B}}$ in both sides of eq. (33), applying the bounds for $A_1$ and $\mathbb{E}_{\mathcal{B}} A_2$ in eqs. (34) and (35), and noting that $\mathbb{E}_{\mathcal{B}} \big\| \widehat{\nabla} g(\theta_t, \delta_t^K; \mathcal{B}) \nabla f(u_{t+1}) \big\|^2 \leq C^2 \mathbb{E}_{\mathcal{B}} \big\| \widehat{\nabla} g(\theta_t, \delta_t^K; \mathcal{B}) \big\|^2 \leq C^4 \left( 1 + \frac{L}{\mu} \right)^2$, we obtain

$$\mathbb{E}_{\mathcal{B}} F(\theta_{t+1}) - F(\theta_t) \leq - \frac{\beta_t}{2} \big\| \nabla F(\theta_t) \big\|^2 + \beta_t \Gamma \Delta_0 (1 - \alpha\mu)^K + \eta_t \mathbb{E}_{\mathcal{B}} \big\| g(\theta_t, \delta_t^K) - u_{t+1} \big\|^2$$

$$+ \frac{\beta_t^2 L^2}{\eta_t} C^2 \left( 1 + \frac{L}{\mu} \right)^2 \big\| \nabla F(\theta_t) \big\|^2 + \frac{L_F \beta_t^2}{2} C^4 \left( 1 + \frac{L}{\mu} \right)^2$$

$$\leq - \beta_t \left( \frac{1}{2} - \frac{\beta_t L^2}{\eta_t} C^2 \left( 1 + \frac{L}{\mu} \right)^2 \right) \big\| \nabla F(\theta_t) \big\|^2 + \beta_t \Gamma \Delta_0 (1 - \alpha\mu)^K$$

$$+ \eta_t \mathbb{E}_{\mathcal{B}} \big\| g(\theta_t, \delta_t^K) - u_{t+1} \big\|^2 + \frac{L_F \beta_t^2}{2} C^4 \left( 1 + \frac{L}{\mu} \right)^2.$$

Then, the proof is complete. $\qquad\square$

### E.2 PROOF OF THEOREM 2 (I.E., THEOREM 1)

Denote $V_t = F(\theta_t) + \big\| g(\theta_{t-1}, \delta_{t-1}^K) - u_t \big\|^2$. Then, using eq. (32) we obtain

$$\mathbb{E}_{\mathcal{B}} V_{t+1} - V_t \leq - \beta_t \alpha_t \big\| \nabla F(\theta_t) \big\|^2 + \beta_t \Gamma \Delta_0 (1 - \alpha\mu)^K - \big\| g(\theta_{t-1}, \delta_{t-1}^K) - u_t \big\|^2$$

$$+ (1 + \eta_t) \mathbb{E}_{\mathcal{B}} \big\| g(\theta_t, \delta_t^K) - u_{t+1} \big\|^2 + \frac{1}{2} L_F \beta_t^2 C^4 \left( 1 + \frac{L}{\mu} \right)^2$$

$$\leq - \beta_t \alpha_t \big\| \nabla F(\theta_t) \big\|^2 + \beta_t \Gamma \Delta_0 (1 - \alpha\mu)^K - \big\| g(\theta_{t-1}, \delta_{t-1}^K) - u_t \big\|^2$$

$$+ (1 + \eta_t)(1 - \eta_t) \big\| g(\theta_{t-1}, \delta_{t-1}^K) - u_t \big\|^2 + \frac{2(1 + \eta_t)}{|\mathcal{B}|} \eta_t^2 \sigma_g^2$$

$$+ \frac{C^2}{\eta_t} (1 + \eta_t)(1 + \kappa^2) \beta_t^2 C^4 \left( 1 + \frac{L}{\mu} \right)^2 + \frac{1}{2} L_F \beta_t^2 C^4 \left( 1 + \frac{L}{\mu} \right)^2, \tag{36}$$

where the last inequality follows from lemma 2. Further, following from the fact that $(1-\eta_t)(1+\eta_t) = 1 - \eta_t^2 < 1$, we obtain

$$\mathbb{E}_{\mathcal{B}} V_{t+1} - V_t \leq - \beta_t \alpha_t \big\| \nabla F(\theta_t) \big\|^2 + \beta_t \Gamma \Delta_0 (1 - \alpha\mu)^K + \frac{2(1 + \eta_t)}{|\mathcal{B}|} \eta_t^2 \sigma_g^2$$

$$+ \frac{1 + \eta_t}{\eta_t} \beta_t^2 C^6 (1 + \kappa^2) \left( 1 + \frac{L}{\mu} \right)^2 + \frac{1}{2} L_F \beta_t^2 C^4 \left( 1 + \frac{L}{\mu} \right)^2. \tag{37}$$

Now, select $\eta_t \in [\frac{1}{2}, 1)$ and $\beta_t$ such that $\alpha_t \geq \frac{1}{4}$, i.e., $\beta_t \leq \frac{1}{2 L_F^2 C^2 \left( 1 + \frac{L}{\mu} \right)^2}$. Hence, taking total expectation of eq. (37) yields

$$\mathbb{E} V_{t+1} - \mathbb{E} V_t \leq - \frac{\beta_t}{4} \mathbb{E} \big\| \nabla F(\theta_t) \big\|^2 + \beta_t \Gamma \Delta_0 (1 - \alpha\mu)^K + \frac{4\sigma_g^2}{|\mathcal{B}|}$$

$$+ 4\beta_t^2 C^6 (1 + \kappa^2) \left(1 + \frac{L}{\mu}\right)^2 + \frac{1}{2} L_F \beta_t^2 C^4 \left(1 + \frac{L}{\mu}\right)^2$$

$$= -\frac{\beta_t}{4} \left\|\nabla F(\theta_t)\right\|^2 + \beta_t \Gamma \Delta_0 (1 - \alpha\mu)^K + \frac{4\sigma_g^2}{|\mathcal{B}|} + \beta_t^2 D_\kappa, \tag{38}$$

where we define $D_\kappa = \left(4C^2(1 + \kappa^2) + \frac{1}{2} L_F\right)(1 + \kappa)^2 C^4$. Therefore, telescoping eq. (38) over $t$ from 0 to $T - 1$ yields

$$\mathbb{E} V_T - V_0 \leq -\sum_{t=0}^{T-1} \frac{\beta_t}{4} \mathbb{E}\left\|\nabla F(\theta_t)\right\|^2 + \frac{4\sigma_g^2 T}{|\mathcal{B}|} + \Gamma \Delta_0 (1 - \alpha\mu)^K \sum_{t=0}^{T-1} \beta_t + D_\kappa \sum_{t=0}^{T-1} \beta_t^2.$$

Thus, rearranging terms, we obtain

$$\frac{\sum_{t=0}^{T-1} \beta_t \mathbb{E}\left\|\nabla F(\theta_t)\right\|^2}{\sum_{t=0}^{T-1} \beta_t} \leq \frac{16\sigma_g^2 T}{|\mathcal{B}| \sum_{t=0}^{T-1} \beta_t} + 4\Gamma \Delta_0 (1 - \alpha\mu)^K + 4D_\kappa \frac{\sum_{t=0}^{T-1} \beta_t^2}{\sum_{t=0}^{T-1} \beta_t} + \frac{4V_0}{\sum_{t=0}^{T-1} \beta_t}. \tag{39}$$

Hence, the proof is complete by choosing the batchsize $|\mathcal{B}| = \mathcal{O}(T)$ and stepsize $\beta_t = \frac{1}{\sqrt{T}}$.

