# OpenReview forum: "Doubly Robust Instance-Reweighted Adversarial Training"
_ICLR.cc/2024/Conference — ICLR 2024 poster_

### Official Review · Reviewer_6nnk · 2023-10-29

**Soundness:** 3 good
**Presentation:** 3 good
**Contribution:** 3 good
**Rating:** 8
**Confidence:** 5

**Summary:**

This paper proposes a novel framework called Doubly Robust Instance-Reweighted Adversarial Training to address the issues of heuristics and non-uniform robust performance in adversarial training. The approach utilizes distributionally robust optimization techniques to obtain importance weights and boost robustness on vulnerable examples. The experiments show that the proposed method outperforms state-of-the-arts on standard classification datasets.

**Strengths:**

1. The proposed framework addresses the issues of heuristics and non-uniform robust performance in adversarial training. The authors use a doubly robust optimization (DRO) approach that is theoretically grounded. It provides a principled way to reweight the training examples based on their vulnerability to adversarial attacks.

2. Even the algorithm falls under the category of iteratively-reweighted adversarial attack, this paper has a more principled optimization formulation than previous works because its DRO approach combines two different models to estimate the importance weights of each training example, and to estimate the importance weights, which is more robust to model misspecification and can handle a wider range of distributional shifts compared to traditional optimization methods. The obtained weights are optimal for the DRO optimization problem defined in Eq. 5 (with the closed-form exact solution for the weights), rather than being ad-hoc picked. This is the most important difference form previous instance-wise or iterative attacks.

3. The bilevel optimization formulation of AT gives one the flexibility to separately design the inner and outer level objectives. This enables the authors to independently construct a new outer level objective that also solves for the instance weights w, and an inner level objective for regularized attack. This flexibility allows for a more generic and powerful framework than the traditional AT formulation, which is limited to a single objective function.

4. The proposed method outperforms several state-of-the-art baselines on standard classification datasets, in terms of robustness against multiple adversarial attacks. They also show that their method can improve the robustness of the weakest (worst-case) data points, which is an important property for real-world applications.

**Weaknesses:**

Since the algorithm requires computing Jacobian inner products to perform parameter updates in the bi-level optimization, could the authors comment on the incurred time complexity? I am wondering if the algorithm runs much slower than vanilla AT (but only improves the robust accuracy moderately).

In their experiments, the authors have compared with AutoAttack which is good, but not with other SOTA methods such as TRADES or Diffusion-based Defense (ICML 2023). Adding some more comparison method would be good.

**Questions:**

See the above.

---

> ### Author Response · Authors · 2023-11-20
> **Response to Reviewer 6nnk**
>
> Thank you for your thorough reviews and constructive comments. We provide our response to your comments below.
>
> Q1: Since the algorithm requires computing Jacobian inner products to perform parameter updates in the bi-level optimization, could the authors comment on the incurred time complexity? I am wondering if the algorithm runs much slower than vanilla AT (but only improves the robust accuracy moderately).
>
> A1: Many thanks for this useful question! We agree that Jacobian-vector product computation in bilevel optimization may cause additional compute burden. However, in terms of complexity it is a known fact from automatic differentiation that computing a Jacobian-vector product has order-wise (up to a constant) the same time and space complexity of gradient computation when reverse-mode automatic differentiation is employed (please see Griewank & Walther, (2008) or  Baydin et al. (2015) for a shorter version).
>
> Moreover, we also conducted additional experiments that use a first-order approximation of our hypergradient estimator (i.e., we ignore the second part of the hypergradient $\nabla L(\theta)$ that contains the second-order computation due to bilevel optimization). We found that our algorithm still maintains the same performance level while achieving exactly the same running-time as standard AT. For example on the CIFAR10, we obtain for the DONE-GD algorithm: 83.13% SA and 57.28% RA-PGD. These new results suggest that the instance weights optimization using DRO is in fact the most important feature of our algorithm. We will include more results about these findings in our revision.
>
> Q2: In their experiments, the authors have compared with AutoAttack which is good, but not with other SOTA methods such as TRADES or Diffusion-based Defense (ICML 2023). Adding some more comparison method would be good.
> A2:  Many thanks for the suggestions! We have conducted new experiments following the most recent SOTA in [1] published in ICML’23 (we assume the reviewer refers to the diffusion-based defense method in [1]). The study in [1] uses a larger CIFAR10 dataset augmented with millions of synthetic images generated by a diffusion model. Note that this falls under a category of new models that boost the performance of adversarial defense using self-supervised learning on tens of millions of additional synthetic images.  With this boosting strategy, our DONE-GD algorithm achieves 86.65% SA and 60.74% AA for the ResNet18 model. As a comparison, [1] achieves 86.42% SA and 58.51% AA. We will include those results in our revision. In our final version, we will also try to add more larger datasets such as one of the imageNet derivatives (tiny imagenet or tiered imagenet).
>
> Usually the TRADES approach is an orthogonal approach and can be used on top of other methods to improve performance against autoAttacks. By the reviewer’s suggestion and for completeness, we will include the comparison with TRADES in our revision.
>
> Reference:
>
> [1] Better Diffusion Models Further Improve Adversarial Training. ICML, 2023.
>
> [Griewank 08’] Griewank, Andreas and Walther, Andrea. Evaluating Derivatives: Principles and Techniques of Algorithmic Differentiation. SIAM, second edition, 2008.
>
> [Baydin 15’] Baydin, Atilim Gunes, Pearlmutter, Barak A., Radul, Alexey Andreyevich, and Siskind, Jeffrey Mark. Automatic differentiation in machine learning: a survey. arXiv preprint arXiv:1502.05767, 2015.
>
> Finally, we thank the reviewer again for the helpful comments and suggestions for our work. If our response clarifies your concerns to a satisfactory level, we kindly ask the reviewer to consider raising the rating of our work. Certainly, we are more than happy to address any further questions that you may have during the discussion period.

---

> > ### Author Response · Authors · 2023-11-22
> >
> > Dear Reviewer,
> >
> > As the author-reviewer discussion period will end soon, we will appreciate it if you could check our response to your review comments. This way, if you have further questions and comments, we can still reply before the author-reviewer discussion period ends. Thank you very much for your time and efforts!

---

> > > ### Comment · Reviewer_6nnk · 2023-11-22
> > > **post rebuttal comments**
> > >
> > > Thank authors for addressing my concerns.
> > > I have no further comments.
> > > I think it is a well-shaped paper, which can be a good addition to the community.
> > > Therefore, I raise my score to vote for accept.

---

> > > > ### Author Response · Authors · 2023-11-22
> > > >
> > > > We thank the reviewer very much for checking our response and kindly increasing the rating. Many thanks for your effort into this review process.

---

### Official Review · Reviewer_bEW4 · 2023-11-01

**Soundness:** 3 good
**Presentation:** 3 good
**Contribution:** 3 good
**Rating:** 6
**Confidence:** 4

**Summary:**

This paper addressed the challenge of adversarial robustness on most vulnerable samples. The existing approaches adopt a instance-reweighted strategy towards improving the worse case robustness. However, there is no principled way to estimate the per-sample weight. This work combines instance-reweighting with bi-level optimization for adversarial robustness. The min-max problem for instance-reweighting optimization was solve with a equivalent compositional bilevel optimization problem.

**Strengths:**

Strength:

1. The mathematical formulation of instance-reweighted bilevel optimization is solved in an elegant manner.

2. The evaluation on imbalanced dataset suggest the worst case adversarial robustness can be improved.

**Weaknesses:**

Weakenss:

1. The improvements on PGD and AutoAttack seem to be less significant. The more significant improvements are observed from RA-Tail-30. Therefore, it is necessary to provide more details of the evaluation protocol for RA-Tail-30.

2. Since the advantage is mainly demonstrated at the imbalanced dataset, the current evaluations on Imbalanced datasets (CIFAR10 and SVHN imbalanced) are not enough for analyzing the performance breakpoint.

3. Comparisons with more recent adversarial training methods are missing.

**Questions:**

It is encouraged to make comparisons with more recent adversarial training methods.

Experiments on more diverse imbalance degrees are necessary for more comprehensive evaluation.

---

> ### Author Response · Authors · 2023-11-20
> **Response to Reviewer bEW4**
>
> Thank you for your thorough reviews and constructive comments. We provide our response to your comments below.
>
> Q1: The improvements on PGD and AutoAttack seem to be less significant. The more significant improvements are observed from RA-Tail-30. Therefore, it is necessary to provide more details of the evaluation protocol for RA-Tail-30.
>
> A1: Many thanks for this useful comment! As we discuss in the introduction, a critical limitation of the conventional AT method is that it suffers a severe non-uniform performance across the empirical distribution. For example, while the average robust performance of the AT method on the CIFAR10 dataset can be as high as 49%, the robust accuracy for the weakest class is as low as 14%, which depicts a huge disparity in robust performance across different classes. So based on this critical remark, we propose the RA-Tail-30 metric which represents the robust performance (i.e., RA-PGD) on the 30% most vulnerable classes, and hence can be seen as a proxy measure of the robustness against attacks on the most vulnerable data points.
>
> Q3: Comparisons with more recent adversarial training methods are missing.
>
> A3: Many thanks for the suggestions! We have conducted new experiments to compare with [1] published in ICML’23, which is the most recent adversarial training SOTA. Note that [1] is a diffusion-based defense method and falls under a category of new models that boost the performance of adversarial defense using self-supervised learning on tens of millions of additional synthetic images. For example [1] uses the latest SOTA diffusion model to generate tens of millions of synthetic images for CIFAR10 to boost the adversarial robustness.
> We've conducted a new experiment using 1 million images generated for CIFAR10 by the same diffusion model as in [1]. Using this, our DONE-GD algorithm achieves 86.65% SA and 60.74% AA for the Resnet18 model. As a comparison, [1] achieves 86.42% SA and 58.51% AA. We will include those results in our final revision.
>
> Q3: Since the advantage is mainly demonstrated at the imbalanced dataset, the current evaluations on Imbalanced datasets (CIFAR10 and SVHN imbalanced) are not enough for analyzing the performance breakpoint. Experiments on more diverse imbalance degrees are necessary for more comprehensive evaluation.
>
> A3: Many thanks for the suggestion! We've conducted new experiments on more diverse imbalance degrees on the CIFAR10 and SVHN datasets. We report the empirical results below.
> | Dataset & Ratio | Method |   SA   | RA-PGD | RA-Tail-30 |
> |:---------------:|:------:|:------:|:------:|:----------:|
> | **CIFAR10 r=0.1**   |   AT   |  61.8  |  41.3  |     3.1    |
> |                 |  ours  | **70.5** | **44.9** |   **11.2**   |
> | **CIFAR10 r=0.2**   |   AT   | 69.74  | 42.37  |    6.25    |
> |                 |  ours  | **74.2** | **48.3** |   **17.2**   |
> | **CIFAR10 r=0.5**   |   AT   |   77   |  47.2  |    16.2    |
> |                 |  ours  | **78.8** | **54.4** |   **27.8**   |
> | **SVHN r=0.1**      |   AT   |  82.5  |  45.4  |    28.4    |
> |                 |  ours  | **82.7** | **48.2** |   **30.6**   |
> | **SVHN r=0.2**      |   AT   |  88.5  |  51.1  |    33.7    |
> |                 |  ours  | **88.9** | **55.9** |   **41.1**   |
> | **SVHN r=0.5**      |   AT   | **91.6**  |  55.2  |    42.7    |
> |                 |  ours  |  91.3  | **62.5** |   **47.9**   |
>
> Again, we thank the reviewer for the suggestions and we will do our best to include even more results related to this aspect in our final version.
>
> Reference:
>
> [1] Better Diffusion Models Further Improve Adversarial Training. ICML, 2023.
>
> Finally, we thank the reviewer again for the helpful comments and suggestions for our work. If our response clarifies your concerns to a satisfactory level, we kindly ask the reviewer to consider raising the rating of our work. Certainly, we are more than happy to address any further questions that you may have during the discussion period.

---

> > ### Author Response · Authors · 2023-11-22
> >
> > Dear Reviewer,
> >
> > As the author-reviewer discussion period will end soon, we will appreciate it if you could check our response to your review comments. This way, if you have further questions and comments, we can still reply before the author-reviewer discussion period ends. Thank you very much for your time and efforts!

---

### Official Review · Reviewer_u17N · 2023-11-01

**Soundness:** 3 good
**Presentation:** 3 good
**Contribution:** 3 good
**Rating:** 6
**Confidence:** 5

**Summary:**

This paper proposes a instance reweighting based adversarial training (AT) framework. Consequently, the authors follow the setting of Zhang et al. 2022  (bilevel optimization formulation for AT) and add the instance reweighting mechanism into it. Moreover, the authors seek to  build a model in the outer level problem that is robust not only to the adversarial examples but also to the worst-case attack distribution.  Compared with the exisiting instance reweighting AT methods, the proposed method  obtain the importance weights by distributionally robust optimization (DRO). The DRO is a more sophisticated choice than the heuristic/geometric schemes of instance rewweighting. Furthermore, the authors propose an equivalent compositional optimization problem (Eq. (6)) and adopt the log-barrier penalty function to drop the challenging $\ell_{\infty}$ norm constraint. The final optimization problem is Eq. (7) and the authors modify SCGD into the compositional implicit differentiation (CID) algorithm to solve it. With some common used assumptions, the authors establish the convegence result for CID.
In the experimental studies, the authors compare three instance re-weighted adversarial training methods with the proposed method on four small-scale datasets. The proposed method show promising improvement on RA-PGD, RA-Tail-30 and RA-AA metric.

**Strengths:**

1. The paper is well-written and easy to follow.
2. The motivation is clear and the equivalent compositional optimization problem is reasonable.
3. The proposed CID method has convergence guarantee.

**Weaknesses:**

1. The empirical studies is not sufficient. Only small-scale datasets is adopted in the experiment.
2. The computational analysis is missing.
3. The justifiability of the assumptions is not discussed.

**Questions:**

1. In Eq. (7)，is the constraint $\delta\in\mathcal{C}_i$ correct? The author claim that "Note that now the constraint $\{\delta\in\mathcal{C}_i\}$ is never binding in Equation (7), because the log-barrier penalty forces the minimizer of $\ell^{bar}_{i}$ to be strictly inside the constraint set." Moreover, in Algorithm 1 Line 5-7, why need the projected operator to keep $\delta_{i,t}^{k}$ in $\mathcal{C}$?

2. It is better to discuss the justifiability  of Assumption 1-3 for AT problem.

3. The SA performance is a weaknness of the proposed method. It is better to explain this limitation.

4. It is better to add some statistical analyses like  P-values, CIs, effect sizes, and so on.

**Details Of Ethics Concerns:**

NA.

---

> ### Author Response · Authors · 2023-11-20
> **Response to Reviewer u17N**
>
> Thank you for your thorough reviews and constructive comments. We provide our response to your comments below.
>
> Q1: In Eq. (7)，is the constraint $\delta \in \mathcal{C}_i$ correct? The author claimed that
> "Note that now the constraint  $\delta \in \mathcal{C}_i  is never binding in eq. (7), because the log-barrier penalty forces the minimizer of $\ell_i^{bar}(\theta, \delta)$ to be strictly inside the constraint set". Moreover, in Algorithm 1 line 5-7, why need the projected operator to keep  $\delta_{i,t}^k \in \mathcal{C}$.
>
> A1: Many thanks for this useful question! Yes, the constraint $\delta \in \mathcal{C}_i$ is correct but \emph{it can be dropped} in eq. (7) because we know that the minimizer of the log-barrier loss $\ell_i^{bar}(\theta, \delta)$ has to be strictly inside the constraint set. In other words, the minimizer of the constrained problem is the same as the non-constrained one (hence we referred to the constraint as never binding).
>
> Further, note that $\hat \delta^*_i(\theta)  in eq. (7) corresponds to the true minimizer of $\ell_i^{bar}(\theta, \delta)$ and we know that it is strictly inside the constraint set. However, the algorithm iterates $\delta_{i,t}^k \in \mathcal{C} (which are approximations of $\hat \delta^*_i(\theta)$) might be outside the constraint set, and hence we need to have the projection operator to push them back into the constraint set if they do fall outside. The main purpose of the log-barrier regularizer is to favor differentiability and guarantee the existence of the implicit gradient.
>
>
>
> Q2: It is better to discuss the justifiability of Assumption 1-3 for AT problem.
>
> A2: Thanks for the suggestion! Assumptions 1-3 are standard assumptions widely adopted in bilevel optimization literature [Ji ‘21; Grazzi ‘20]. Specifically, Assumption 3 characterizes the variance of cost function over data samples, which is typically bounded. Assumptions 1-2 require the Lipschitzness of the objective function and its first- and second-order derivatives, which is crucial for analyzing convergence performance of bilevel optimization.
>
>
> Q3: The SA performance is a weakness of the proposed method. It is better to explain this limitation.
>
> A3: This is indeed an important comment! Yes, while our approach generally improves the robust performance, it did not improve the SA (but it still has comparable SA performance with the other baselines). In order to further investigate this in terms of the overall performance (i.e., SA-RA tradeoff) of our approach, we have run new experiments with evaluations on 70% clean-30% adversarial, 50% clean-50% adversarial; 30% clean-70% adversarial; and all are randomly i.i.d. chosen. This captures the practical real world scenarios where only fractions of the input data are attacked. We provide the experimental results for the CIFAR10 dataset below.  Note that Xc-Ya means performance on  X% clean and Y% adversarial.
>
> | method | 0c-100a | 30c-70a | 50c-50a | 70c-30a| 100c-0a |
> | --- | --- | --- | --- | -- | -- |
> | Ours | 57.46 | 65.41 | 70.15 | 75.38 | 83.41|
> | AT | 49.29 | 59.33 | 66.02 | 72.29 | 82.1 |
>
> We found that our method is still able to interpolate on the good decent standard and robust accuracy. We thank the reviewer for bringing this up and will include those discussions in our final revision.
>
> Q4: The computational analysis is missing.
>
> A4: Thanks for the comment! Compared to the conventional AT method, only the Jacobian-vector product computation due to bilevel optimization may cause additional compute burden. However, in terms of time and memory complexity it is a known fact from automatic differentiation that computing a Jacobian-vector product has order-wise (up to a constant) the same time and space complexity of gradient computation when reverse-mode automatic differentiation is employed (please see [Griewank 08’] or  [Baydin 15’] for a shorter version).
>
> Moreover, we also conducted additional experiments that use a first-order approximation of our hypergradient estimator (i.e., we ignore the second part of the hypergradient $\nabla L(\theta)$ that contains the second-order computation due to bilevel optimization). We found that our algorithm still maintains the same performance level while achieving exactly the same running-time as standard AT. For example on the CIFAR10, we obtain for the DONE-GD algorithm: 83.13% SA and 57.28% RA-PGD. These new results suggest that the instance weights optimization using DRO is in fact the most important feature of our algorithm. We will include more results about these findings in our revision.

---

> ### Author Response · Authors · 2023-11-20
> **Response to Reviewer u17N continued**
>
> Q5: ​​The empirical studies is not sufficient. Only small-scale datasets is adopted in the experiment.
>
> A5: Many thanks for the suggestions! We have conducted new experiments following the most recent SOTA in [1] published in ICML’23, which uses a larger CIFAR10 dataset augmented with 1 million synthetic images generated by a diffusion model. Note that this falls under a category of new models that boost the performance of adversarial defense using self-supervised learning on tens of millions of additional synthetic images.  With this boosting strategy, our DONE-GD algorithm achieves 86.65% SA and 60.74% AA for the ResNet18 model. As a comparison, [1] achieves 86.42% SA and 58.51% AA. We will include those results in our revision. In our final version, we will also try to add more larger datasets such as one of the imageNet derivatives (tiny imagenet or tiered imagenet).
>
>
>
> Q6: It is better to add some statistical analyses like P-values, CIs, effect sizes, and so on.
>
> A6: We thank the reviewer for the useful suggestions! We’ve run the CIFAR10 experiments 10 times for our method and the AT method and used the t-test to compute the p-value for each of the metrics SA, RA-PGD, and RA-Tail-30.
>
> | Metric     | Our        | AT        | P-value        |
> |------------|-------------------|------------------|----------------|
> | SA         | 83.45 (±0.51)     | 82.86 (±0.47)    | 0.039          |
> | RA-PGD     | 57.69 (±0.36)     | 48.96 (±0.49)    | 1.525e-13      |
> | RA-Tail-30 | 40.22 (±0.43)     | 28.20 (±0.31)    | 1.536e-14      |
>
> Note that the p-value measures the significance of the advantage of our method compared to AT. The lower the p-value, the higher significance of our advantage over AT.
> Clearly in the above table, these results (all p-values less than 0.05) show that our method has a meaningful and consistent performance advantage over the AT method across these metrics for this dataset. We will do our best to include more statistical metrics in our final revision.
>
> References:
>
> [1] Better Diffusion Models Further Improve Adversarial Training. ICML, 2023.
>
> [Griewank 08’] Griewank, Andreas and Walther, Andrea. Evaluating Derivatives: Principles and Techniques of Algorithmic Differentiation. SIAM, second edition, 2008.
>
> [Baydin 15’] Baydin, Atilim Gunes, Pearlmutter, Barak A., Radul, Alexey Andreyevich, and Siskind, Jeffrey Mark. Automatic differentiation in machine learning: a survey. arXiv preprint arXiv:1502.05767, 2015.
>
> [Grazzi 20’] Riccardo Grazzi, Luca Franceschi, Massimiliano Pontil, and Saverio Salzo. On the iteration complexity of hypergradient computation. ICML, 2020.
>
> [Ji 21’] ​​Kayi Ji, Junjie Yang, and Yingbin Liang. Bilevel optimization: Convergence analysis and enhanced design. ICML, 2021.
>
> Finally, we thank the reviewer again for the helpful comments and suggestions for our work. If our response clarifies your concerns to a satisfactory level, we kindly ask the reviewer to consider raising the rating of our work. Certainly, we are more than happy to address any further questions that you may have during the discussion period.

---

> > ### Author Response · Authors · 2023-11-22
> >
> > Dear Reviewer,
> >
> > As the author-reviewer discussion period will end soon, we will appreciate it if you could check our response to your review comments. This way, if you have further questions and comments, we can still reply before the author-reviewer discussion period ends. Thank you very much for your time and efforts!

---

### Meta-Review · Area_Chair_C9rY · 2023-12-05

**Metareview:**

This paper proposes a novel instance reweighting adversarial training (AT) framework that incorporates distributionally robust optimization (DRO) to stress the issue of adversarial robustness in the most vulnerable samples. The reviewers generally find the paper well-written, with a clear motivation and an elegant solution to the instance-reweighted bi-level optimization problem. The method's strength lies in its progressive approach to generating perturbed data and the convergence guarantee of the proposed CID method. Notably, the proposed method shows promising improvement over traditional adversarial training, particularly in the context of imbalanced datasets.

However, reviewers point out the experimental studies are limited to small-scale datasets, and that the authors should include comparisons with more recent adversarial training methods and expand the experimental protocol for more diverse imbalance degrees. They also raised concerns about the potential increase in time complexity due to the computation of Jacobian inner products. The authors provided additional experimental results by comparing with Diffusion Defense (ICML 2023) on SVHN and CIFAR-10.

Overall, while the reviewers appreciate the novelty and the theoretical grounding of the DRO approach, the relative small-scale empirical studies have limited the enthusiasm for endorsing the paper. The current consensus is still supportive of acceptance, provided that the authors can address the mentioned concerns and bolster their empirical evidence.

**Justification For Why Not Higher Score:**

the experimental studies are limited to small-scale datasets, and that the authors should include comparisons with more recent adversarial training methods and expand the experimental protocol for more diverse imbalance degrees. They also raised concerns about the potential increase in time complexity due to the computation of Jacobian inner products. The authors provided additional experimental results by comparing with Diffusion Defense (ICML 2023) on SVHN and CIFAR-10.

**Justification For Why Not Lower Score:**

The method's strength lies in its progressive approach to generate perturbed data and the convergence guarantee of the proposed CID method. Notably, the proposed method shows promising improvement over traditional adversarial training, particularly in the context of imbalanced datasets.

---

### Decision · Program_Chairs · 2024-01-16

Accept (poster)